# Metabolic requirements of NK cells during the acute response against retroviral infection

Elisabeth Littwitz-Salomon [1✉], Diana Moreira[1], Joe N. Frost[2], Chloe Choi[1], Kevin T. Liou [3], David K. Ahern[2], Simon O'Shaughnessy[1], Bernd Wagner[4], Christine A. Biron[3], Hal Drakesmith [2], Ulf Dittmer[5] & David K. Finlay [1,6✉]

Natural killer (NK) cells are important early responders against viral infections. Changes in metabolism are crucial to fuel NK cell responses, and altered metabolism is linked to NK cell dysfunction in obesity and cancer. However, very little is known about the metabolic requirements of NK cells during acute retroviral infection and their importance for antiviral immunity. Here, using the Friend retrovirus mouse model, we show that following infection NK cells increase nutrient uptake, including amino acids and iron, and reprogram their metabolic machinery by increasing glycolysis and mitochondrial metabolism. Specific deletion of the amino acid transporter Slc7a5 has only discrete effects on NK cells, but iron deficiency profoundly impaires NK cell antiviral functions, leading to increased viral loads. Our study thus shows the requirement of nutrients and metabolism for the antiviral activity of NK cells, and has important implications for viral infections associated with altered iron levels such as HIV and SARS-CoV-2.

[1] School of Biochemistry and Immunology, Trinity Biomedical Sciences Institute, Trinity College Dublin, 152-160 Pearse Street, Dublin 2, Ireland. [2] MRC Human Immunology Unit, MRC Weatherall, Institute of Molecular Medicine, John Radcliffe Hospital, University of Oxford, Oxford, UK. [3] Department of Molecular Microbiology and Immunology, Brown University, Box G-B171 Meeting Street, Providence, RI 02912, USA. [4] Department of Clinical Chemistry, University Hospital Essen, Essen, Germany. [5] Institute for Virology, University Hospital Essen, University of Duisburg-Essen, Essen, Germany. [6] School of Pharmacy and Pharmaceutical Sciences, Trinity Biomedical Sciences Institute, Trinity College Dublin, 152-160 Pearse Street, Dublin 2, Ireland.
✉email: Elisabeth.Littwitz@uni-due.de; finlayd@tcd.ie

The best protection we have against viruses, such as polio, measles, and hepatitis A or B, are vaccines but the development of such vaccines takes time and in the case of the human immunodeficiency virus (HIV) an effective vaccine is still illusive. Natural killer (NK) cells are cytotoxic lymphocytes that have an important role in the early response against likely all viral infections[1]. Understanding how NK cells contribute to the initial immunity against retroviral infection has the potential to reveal novel strategies for antiretroviral therapy.

NK cells express an array of activating and inhibitory germline-encoded receptors and become activated by the integration of its receptor signals[2]. Activated NK cells eliminate target cells via the release of cytotoxic granules filled with granzymes and perforin and also express death receptor ligands (e.g. TRAIL, FasL), which can induce apoptosis in target cells after binding to its receptors[1]. Upon activation, NK cells also produce a large array of cytokines including Interferon (IFN) γ and Tumour necrosis factor (TNF). NK cells also become activated in response to cytokines including interleukin (IL)-2, IL-12, IL-15 and IL-18, and this is an important mode of NK cell activation during viral and retroviral infections[3–5].

Recent research has demonstrated that once activated by cytokines, NK cells undergo distinct metabolic rearrangements that are essential for their effector functions including cytotoxicity and cytokine production[6–8]. This metabolic response involves the upregulation of nutrient transporters and metabolic enzymes and an increase in mitochondrial mass. Collectively, this facilitates increased flux through metabolic pathways including glycolysis and oxidative phosphorylation (OXPHOS) that allows for the production of energy in the form of ATP and an enhanced biosynthetic capacity. The transcription factor cMyc has been identified as a key regulator of these metabolic changes in activated NK cells[7]. Cytokine-activated NK cells are highly dependent on amino acids availability and are particularly reliant on the flux through the large neutral amino acid transporter Slc7a5, which supports metabolic signal transduction pathways including the protein expression of cMyc[7]. Iron is very important for essential cellular activities including mitochondrial function, DNA repair and synthesis, and epigenetic regulation and sensing of hypoxia[9]. In T cells, there is strong evidence that cMyc is important for the expression of the transferrin receptor (CD71), which is crucial for the uptake of ferric iron into cells[10,11] and it was recently demonstrated that adaptive T cell immunity is critically dependent on serum iron availability[12]. The importance of iron for the function of innate lymphocytes is unknown, however concomitant with increased mitochondrial mass and respiration rates, activated NK cells increase the expression of CD71[7,8]. Perturbations in NK cell metabolism have been described in multiple diseases including cancer and obesity and linked to impaired functional NK cell responses[13]. In the case of obesity, NK cells completely fail to increase levels of cellular metabolism in response to cytokines and have severely impaired cytotoxicity[13]. To date, the metabolic changes that occur in NK cells as they respond to acute retroviral infection and their importance in the overall control of retroviral burden have not been investigated.

Here, we used the Friend retrovirus (FV) mouse model to analyse the NK cell metabolism and NK cell functions after an acute retrovirus infection of mice. Friend virus complex consists of a replication-competent, but apathogenic F-MuLV (Friend Murine Leukaemia Virus) and the replication-defective but pathogenic SFFV (Spleen Focus-Forming Virus)[14]. Adult C57BL/6 mice are resistant to FV-induced disease and develop an early and strong immune response that controls viral replication, but fails to clear infection resulting in a lifelong chronic infection. In retrovirus infections, such as infections with FV, simian immunodeficiency virus (SIV) and HIV, NK cells play an important role in restricting virus replication in the acute phase[15]. Depleting NK cells in mice with acute FV infection has detrimental effects on the control of viral burden[16]. For NK cells activation the cytokines IL-15 and IL-18, produced by Dendritic cells (DCs) and macrophages, are necessary to limit the spread of FV[4]. Previous work has demonstrated that the antiretroviral activity of NK cells can be reduced by regulatory T cells through competition for IL-2 and the IL-10 production[17,18]. NK cells become dysfunctional during chronic infection but a small population of antigen-specific memory-like NK cells develop in retroviral infections as it was demonstrated for SIV and FV[15,19,20]. Thus, the FV infection is a good experimental model when analysing NK cells and metabolic requirements for their responses against retroviruses. In chronically HIV-infected individuals, co-infected with cytomegalovirus (CMV), NK cells show evidence of mitochondria dysfunction and reduced levels of OXPHOS[21]. Targeting immune cell metabolism is an extremely promising therapeutic approach that has been investigated for numerous immunological diseases[22–24]. However, almost nothing is known about the NK cell metabolic requirements and alterations in the acute phase of retroviral infection and so potential new therapies could be missed. The FV mouse model enables the detailed analysis of the metabolic changes that accompany early antiviral NK cell responses against retrovirus infection. The importance of NK cell metabolic processes for these antiviral responses can also be investigated following manipulation of metabolism in vivo.

Here, we show that NK cells reprogramme their metabolism by increasing their nutrient uptake, glycolysis and mitochondrial machinery after acute virus infection. The antiviral functions of NK cells strongly depend on sufficient levels of iron. This new understanding of the metabolism of NK cells in the acute phase of infection is essential to facilitate the development of novel metabolism-targeted approaches for treating infectious diseases.

## Results

**Activated, cytokine-producing NK cells in FV-infected mice.** C57BL/6 mice were infected with 40,000 Spleen Focus-Forming Units (SFFU) of FV and the viral loads in the bone marrow and spleens were monitored over the course of 28 days. The bone marrow and spleen were analysed due to high viral replication in these organs[25]. FV preferentially infects erythroblasts, monocytes and macrophages but all dividing cells in the spleen and bone marrow can be targets for infection[26,27]. As shown in Fig. 1a viral loads increased and peaked at 7 days post-infection (dpi) in both analysed organs. Viral loads were significantly higher in the bone marrow in comparison to the spleen. The viral burden was reduced at 12 dpi, correlating with the activation of the adaptive immune response against FV[28]. FV was not eradicated but was found to persist at 28 dpi and harbouring infected cells in low levels in bone marrow and spleen. To study the role of NK cells in FV immunity, we analysed the activation of NK cells during the course of infection[15]. The NK cell gating strategy is shown in Supplementary Fig. 1d. Activated NK cells express the early activation marker CD69 and can be analysed via flow cytometry. Activation of NK cells reached its maximum at 7 dpi and decreased at 12 dpi and 28 dpi in both organs (Fig. 1b). We further studied NK cells at the peak of the NK cell response (7 dpi), evaluating the NK cell maturation status by measuring the expression of CD11b and CD27 on the cell surface[29]. These molecules classify four stages of NK cells maturation: Double negative (CD27⁻CD11b⁻ immature NK cells), CD27⁺CD11b⁻ (immature cytokine producers), double-positive (CD27⁺CD11b⁺ mature cytokine producer) and CD27⁻ CD11b⁺ (cytotoxic

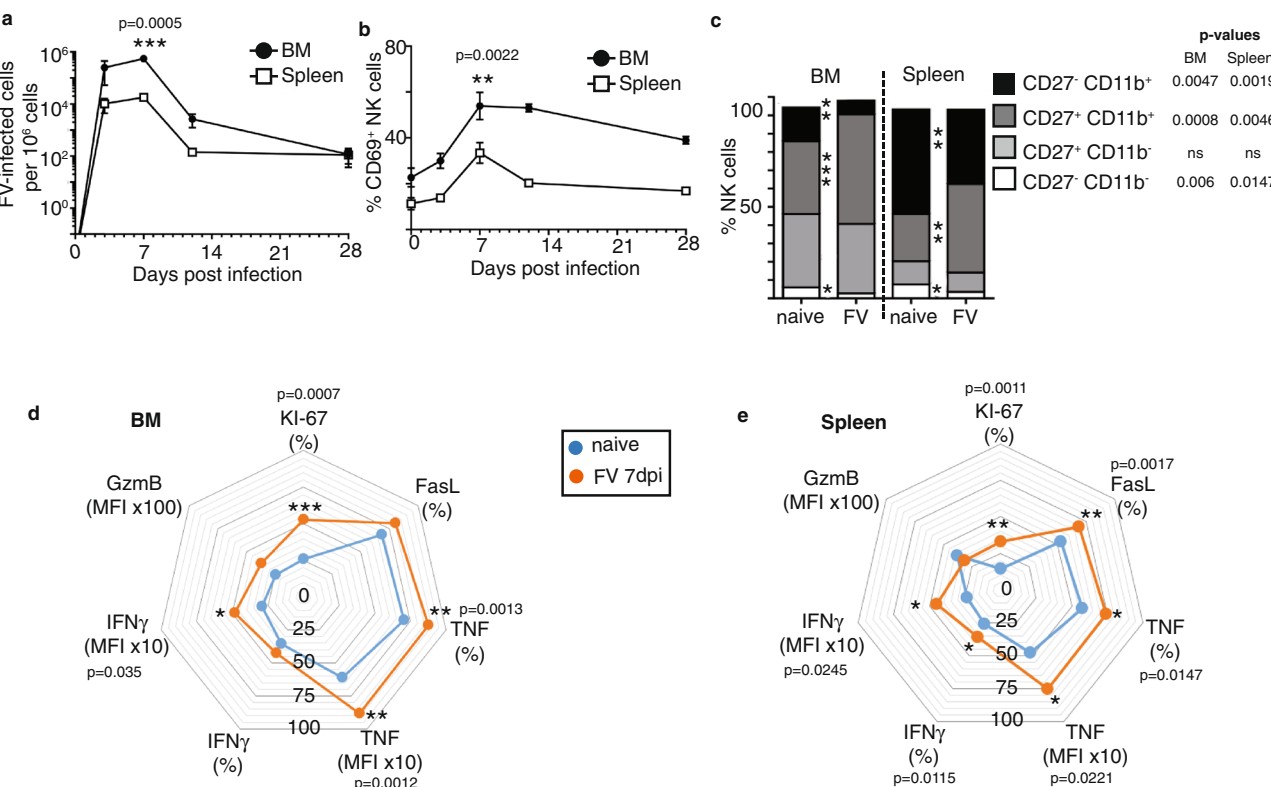

**Fig. 1 Kinetic of FV infection and effector phenotype of NK cells upon FV infection.** C57BL/6 mice were infected with FV for 3, 7, 12 and 28 days. Single-cell suspensions from spleens and bone marrow (BM) were prepared and used for the analysis of viral loads via Infectious Center assay (**a**). At least six mice from at least two individual experiments were used for the analysis. **b** Single-cell suspensions were stained for NK cell markers (CD3$^-$ NK1.1$^+$ CD49b$^+$) and analysed for the activation by measuring the early activation marker CD69. Naive mice were used as control. Mean values ± SEM were indicated by circles (bone marrow) and rectangles (spleen). Statistically significant differences between bone marrow and spleen (**a**) and CD69$^+$ NK cells (**b**) were analysed by Mann–Whitney test. At 7 dpi, splenocytes were stained for NK cell markers and CD27 and CD11b (**c**). The effector phenotype of splenic NK cells (**e**) and NK cells from the bone marrow (**d**), KI-67, FasL, TNF, IFNγ and GzmB is displayed as spider plots. Data of NK cells from naive mice were displayed in blue and from FV infection in red. Statistically significant differences were analysed between naive and FV groups with an unpaired *t*-test (CD27 CD11b, KI-67, FasL, TNF (%), IFNγ (%)) or Mann–Whitney test (TNF (MFI), IFNγ (MFI)) within the bone marrow or spleens. A minimum of six mice from two independent experiments was used for the analysis. Significances are indicated as follows: *$p$ < 0.05, **$p$ < 0.01, ***$p$ < 0.001. Applied statistical tests were two-sided. Source data are provided as a Source Data file. ns not significant.

effector cells). At 7 dpi, the subpopulation of effector cells (CD27$^-$CD11b$^+$), as well as the subpopulation of immature NK cells (CD27$^-$CD11b$^-$), decreased (Fig. 1c). At the same time, we found a higher percentage of CD27$^+$CD11b$^+$ NK cells (mature cytokine producer) in both organs in comparison to NK cells from naive hosts (Fig. 1c). In line with these changes in subset frequencies, we detected an increased production of IFNγ and TNF and more cytokine positive NK cells in bone marrow (Fig. 1d) and spleen (Fig. 1e) at 7 dpi but similar levels of granzyme B expression (Fig. 1d, e). At 7 dpi, we analysed the NK cells positive for CD69, IFNγ, TNF or granzyme B for their subset maturation stage, again using CD27 and CD11b (Supplementary Fig. 1a). Most of the activated NK cells belong to the double-positive subset. Cytokines were produced mainly by the cytokine-producing subsets whereas granzyme B was expressed by mature cytokine producers and cytotoxic effector cells. NK cells from FV-infected mice were also found to express higher levels of the death receptor ligand FasL and KI-67, a marker of proliferating cells compared to naive cells (Fig. 1d, e). These results demonstrate that NK cells are highly activated and increase their cytokine production during acute FV infection.

**FV infection induces metabolically active NK cells.** Cellular growth, cell division and cytokine production are energetically demanding processes that require increased uptake of nutrients

and a reprogramming of the metabolic machinery[8]. Considering that NK cells showed peak activation at 7 dpi, we predicted that this would be associated with metabolic changes in these NK cells. Thus, we addressed the question of whether acute FV infection induces metabolic changes in NK cells. We first analysed whether there was a change in cell size in the activated NK cells following FV infection through analysis of the forward scatter (FSC), as a measure of cell size, by flow cytometry (Fig. 2a, b). There was a clear shift in the FSC for NK cells from spleen and bone marrow upon FV infection, indicative of increased cell size. This increase in NK cell FSC correlated with NK cell activation, as measured by CD69 expression, suggesting that increased cell growth was associated with NK cell activation in FV-infected mice (Fig. 2c). To understand the processes underlying the apparent increase in the size of activated NK cells, we stratified the NK cells from FV-infected mice into FSC$^{high}$ and FSC$^{low}$ populations and analysed the expression of important nutrient transporters. CD98 is a protein complex of a number of transporters for large neutral amino acids including (LAT1), where CD98 complexes with the pore-forming subunit Slc7a5. The transferrin receptor CD71 is an integral membrane glycoprotein important for the uptake of transferrin/iron. CD98 and CD71 expression was significantly increased on the FSC$^{high}$ NK cell population from FV-infected mice (Fig. 2d, e). These increases in CD71 and CD98 expression were not simply because the NK cells were bigger as the

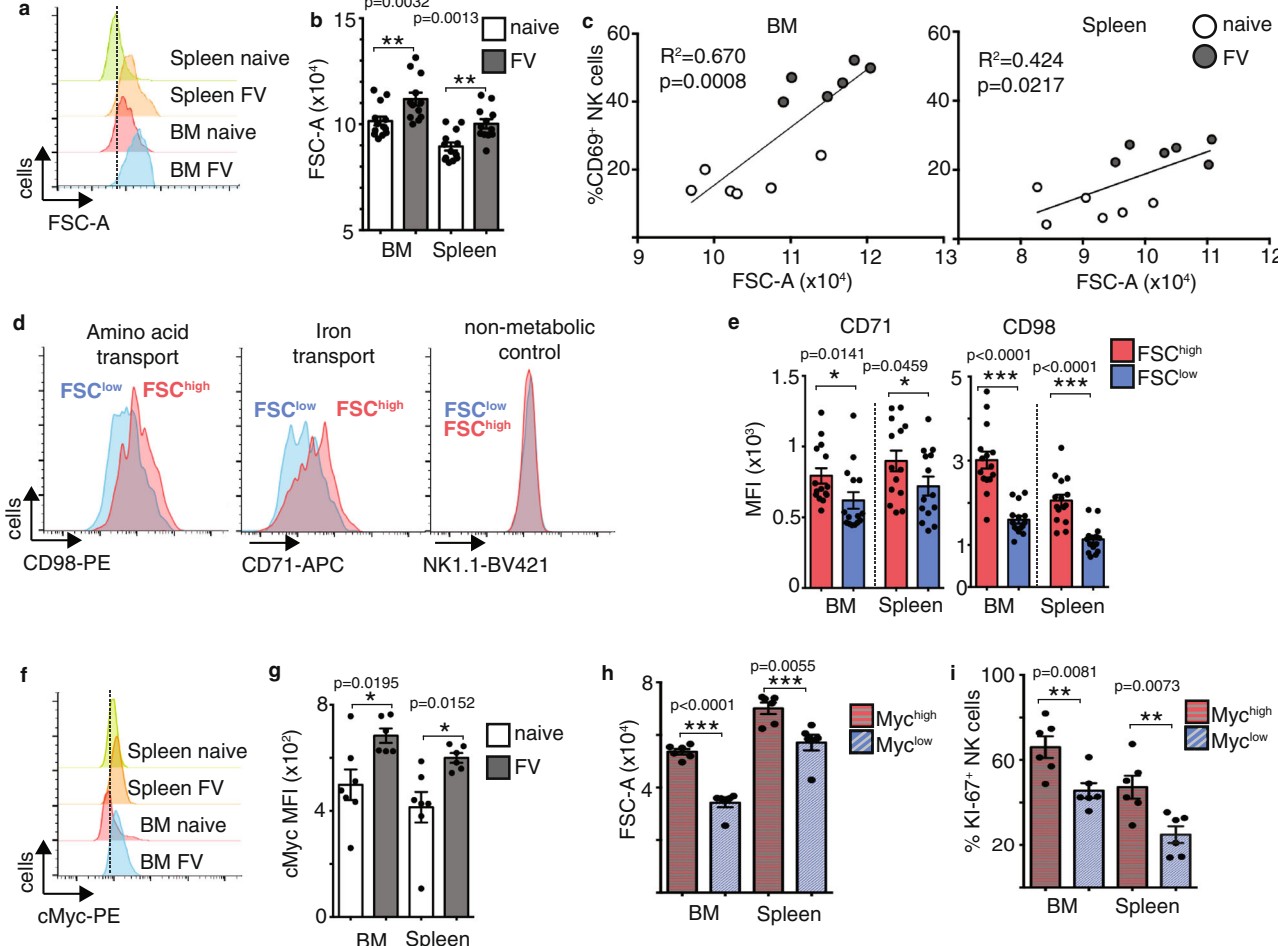

**Fig. 2 Increased NK cell blastogenesis and augmented metabolic activity after acute FV infection.** Splenic and bone marrow (BM) NK cells from naive mice or mice FV-infected for 7 days were analysed for cell size by analysing the FSC on NK cells. NK cells were gated on lymphocytes, single cells, viable, CD3[−], NK1.1[+] CD49b[+] cells. A representative histogram of the FSC on NK cells from both organs are shown in **a** and displayed as a bar graph ± SEM in **b**. Statistically significant differences were analysed between naive and FV groups by a two-tailed unpaired *t*-test within the bone marrow or spleens. Experiments were repeated independently twice with similar results. The correlation between activated NK cells and NK cell size (FSC-A) was analysed and displayed in **c**. NK cells from naive mice are displayed in open circles whereas NK cells from FV-infected mice are displayed in grey circles. At least six mice per group from two independent experiments were used. FSC[high] and FSC[low] NK cells were analysed for the expression of CD98 and CD71 (**d**). NK1.1 expression was analysed on FSC[high] and FSC[low] NK cell population (**d**, right-hand side). Experiments were repeated independently twice with similar results. Differences between FSC[high] and FSC[low] NK cells regarding CD71 and CD98 are shown in a bar graph in **e**. At least 14 mice from four independent experiments were used for the analysis. Statistically significant differences between FSC[high] and FSC[low] NK cells were analysed with a two-tailed Mann–Whitney test within the bone marrow or spleens. Data are presented as mean values ± SEM. Representative histogram of cMyc of NK cells in the spleen and bone marrow are shown in f. Experiments were repeated independently twice with similar results. The MFI of cMyc on NK cells (**g**) is displayed as bar graphs ± SEM and were analysed by a two-tailed unpaired *t*-test. At least six animals from two independent experiments were used for the analysis. cMyc[+] and cMyc[−] NK cells were further analysed for the cell size by measuring the FSC (**h**) and for proliferation by detecting KI-67 (**i**). At least six mice per group from two independent experiments were used. Data are presented as mean values ± SEM. Statistically significant differences in bone marrow or spleen were analysed with a two-sided unpaired *t*-test and displayed as *$p < 0.05$, **$p < 0.01$, ***$p < 0.001$. Source data are provided as a Source Data file.

expression proteins with no function in metabolism, including NK1.1, were equivalent between FSC[high] and FSC[low] NK cell populations (Fig. 2d). In naive control mice, the proportion of FSC[high] NK cells is very small. CD71 is a target of cMyc, which is an important general regulator of cellular metabolism and proliferation. It has been shown that cMyc is essential for the metabolic response of NK cells following their activation by cytokines[7,30]. cMyc can promote growth and proliferation through regulating a number of metabolic processes including glycolysis and the pentose phosphate pathway, a pathway branching off from glycolysis[31]. cMyc also promotes mitogenesis and OXPHOS in activated NK cells[7]. Together these pathways support the generation of ATP and the production of the

biosynthetic molecule for the generation of protein, lipid and nucleotides. Indeed, cMyc expression was significantly increased in NK cells from FV-infected mice compared to naive mice in both bone marrow and spleen (Fig. 2f, g). cMyc was mainly expressed by the CD27[+] NK cell subsets as is shown in Supplementary Fig. 1b. NK cells with high cMyc expression were found to be larger (FSC-A) and more proliferative (KI-67 staining) than NK cells with lower levels of cMyc expression, fitting with the described roles for cMyc in the control of cellular growth and proliferation (Fig. 2h, i). Taken together, NK cells upregulate the metabolic machinery after FV infection, which is important to support the metabolic demands of increased cell growth and proliferation during infection.

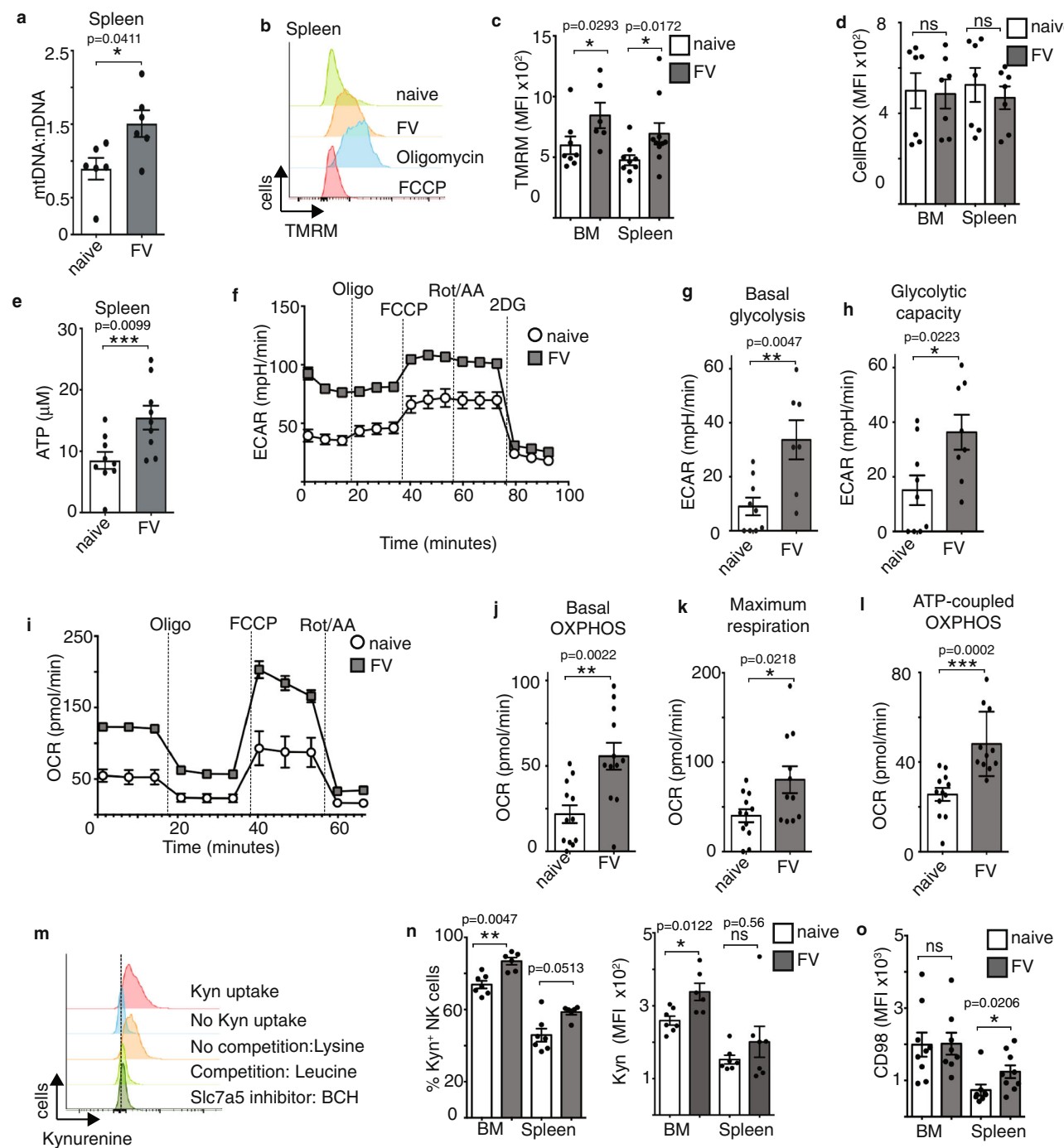

**FV infection increases NK cell mitochondrial capacity**. In cytokine-stimulated NK cells, the transcription factor cMyc is required to facilitate increased glycolytic and mitochondrial metabolism including cytokine-induced increases in mitochondrial mass[7]. Mitochondria generate chemical energy by converting nutrients and oxygen into adenosine triphosphate (ATP), necessary for all biochemical reactions. Here, we considered whether the observed increases in cMyc expression in NK cells following FV infection was linked to changes in mitochondrial mass, mitochondrial fitness and ATP levels. First, we analysed the ratio of the mitochondrial and nuclear DNA of isolated splenic NK cells from naive and FV-infected mice (7 dpi) to obtain information about the mitochondrial mass. After infection of mice with FV, the mt/nDNA ratio was significantly increased, indicating increased

mitochondrial mass (Fig. 3a). Next, we analysed the mitochondrial membrane potential by analysing Tetramethylrhodamine, methyl ester (TMRM) staining (Fig. 3b, c). Oligomycin, which inhibits the ATP synthase, was used as a positive control as it induces the maximal polarisation of the mitochondrial membrane. FCCP, a mitochondrial membrane uncoupler, was used as a negative control as it depolarises the mitochondrial membrane. As shown in Fig. 3b, c, the mitochondrial membrane potential was significantly increased in NK cells after FV infection in comparison to NK cells from naive C57BL/6 mice in both analysed organs. Interestingly, the larger NK cells (FSC^high NK cells) had increased mitochondrial potential and mitochondrial mass in comparison to the smaller NK cells (FSC^low NK cells) from infected mice, further supporting the argument that these larger NK cells have a

**Fig. 3 Increased mitochondrial activity, glycolysis and kynurenine uptake of NK cells in acute FV infection.** C57BL/6 mice were infected with FV and analysed at 7 dpi. Naive C57BL/6 mice were used as control. NK cells were isolated from the spleen using magnetic beads. Mitochondrial and nuclear DNA was detected by quantitative real-time PCR (**a**). Samples were collected from two independent experiments and were run in duplicates or triplicates. A minimum of six mice per group was used and analysed by Mann–Whitney test. Data are presented as mean values ± SEM. Spleen and bone marrow (BM) was harvested at 7 dpi and NK cells were analysed for TMRM (**b**, **c**) and CellROX (**d**). Carbonyl cyanide-4-(trifluoromethoxy)phenylhydrazone (FCCP, minimum) and Oligomycin (maximum) were used as a control for the TMRM measurement. A minimum of six (**c**) or seven (**d**) mice per group from two independent experiments with similar results were used and analysed by Mann–Whitney test. ATP levels were detected in bead-isolated, splenic NK cells from naive and FV-infected mice (**e**). Purified NK cells were lysed with Glo lysis buffer and analysed with an ATP determination kit. Samples were run at least in duplicates and a minimum of nine mice per group from three independent experiments was used. Statistically significant differences were analysed by an unpaired *t*-test. Data are presented as mean values ± SEM for **c**–**e**. Analysis of extracellular acidification rate (ECAR) was measured in isolated, splenic NK cells in naive and FV-infected mice (**f**). Experiments were repeated independently three times with similar results. Basal glycolysis (**g**) and glycolytic capacity (**h**) were calculated and displayed as bar graph ± SEM. At least seven mice from three independent experiments were used and analysed by an unpaired *t*-test. Analysis of oxygen consumption rate (OCR) was measured in isolated, splenic NK cells in naive and FV-infected mice (**i**). Experiments were repeated independently three times with similar results. Basal OXPHOS (**j**), maximum respiration (**k**) and ATP-linked respiration (**l**) was calculated and displayed as bar graph ± SEM. At least twelve mice from four independent experiments were used and analysed by an unpaired *t*-test. A representative histogram of kynurenine (kyn) uptake by splenic NK cells with suitable controls (no kynurenine, lysine, leucine, 2-Amino-2-norbornanecarboxylic acid (BCH)) (**m**) and the quantification is shown for the bone marrow and spleen (**m**, **n**). Experiments were repeated independently twice with similar results. The expression of CD98 by NK cells is shown in **o** ±SEM. A minimum of six (**n**) or eight (**o**) mice per group from two (**n**) or three (**o**) independent experiments were used and analysed by Mann–Whitney test for and displayed ± SEM. *$p < 0.05$, **$p < 0.01$, ***$p < 0.001$. The applied statistical tests were two-sided. Source data are provided as a Source Data file. ns not significant.

metabolically active phenotype (Supplementary Fig. 1c). We also measured the cellular ROS levels in splenic and bone marrow NK cells and found no differences between the naive and infected groups (Fig. 3d). Considering that increased ROS levels can be associated with mitochondrial dysfunction, which has been observed for NK cells in certain pathological situations, we interpreted these normal ROS levels to suggest that NK cells during acute FV infection have healthy mitochondria. Indeed, this was further supported by the fact that NK cells from FV-infected mice had higher concentrations of ATP in comparison to NK cells from naive mice (Fig. 3e). For cytokine-activated NK cells, it has been shown previously that NK cells engage aerobic glycolysis plus high rates of oxidative phosphorylation (OXPHOS), a metabolic configuration that supports ATP production and provision of building blocks for biosynthesis[6,8]. Thus, we determined the extracellular acidification rates (ECAR) of splenic NK cells isolated ex vivo from naive and virus-infected mice (Fig. 3f). NK cells from infected mice exhibited increased ECAR rates in comparison to NK cells from naive mice. The calculation of basal glycolysis (Fig. 3g) and glycolytic capacity (Fig. 3h) confirmed the augmented glycolytic activity in NK cells from FV-infected animals. We also analysed the oxygen consumption rate (OCR) in splenic NK cells isolated from naive or virus-infected mice. As shown in Fig. 3i, j the basal OCR rate was significantly increased in NK cells from FV-infected mice in comparison to naive mice. NK cells from virus-infected mice also had enhanced rates of maximal respiration (Fig. 3k) and increased levels of ATP-linked respiration (Fig. 3l). The current data indicates that NK cells show increased cellular metabolism and elevated levels of the metabolic regulator cMyc during acute FV infection. Amino acid supply is crucial for metabolically active cells to support protein synthesis and other biosynthetic processes. Indeed, the uptake of amino acids through the system L-amino acid transporter has been shown to be important in NK cells stimulated with cytokines in vitro because it sustains elevated levels of cMyc protein[7]. This is because cMyc protein levels are balanced by rapid protein synthesis and rapid degradation via the ubiquitin-proteasome pathway[32]. Thus, in NK cells cMyc levels are acutely sensitive to amino acid availability[7]. Here, we assessed whether FV infection and increased cMyc expression were associated with increased transport of amino acids through system L-amino acid transporters in NK cells. There are four system L-amino acid transporters (LAT1-4) and in addition to large amino acids, they also transport

kynurenine into the cell, which is naturally fluorescent and can be used to assess system L-amino acid transport activity by flow cytometry[33]. Thus, the analysis of kynurenine serves as a measure of amino acid uptake through the Slc7a5 transporter. System L-amino acid transporter consists of a common heavy chain subunit, CD98, and a light chain subunit that mediates the amino acid transport. In vitro cytokine-activated NK cells express only the light chain SLC7A5 (LAT1), but not SLC7A8 (LAT2), SLC43A1 (LAT3), SLC43A2 (LAT4), and so this kynurenine uptake assay serves as a measure of Slc7a5 amino acid uptake capacity[7]. Important controls including inhibitors of Slc7a5 and competition with other transported amino acids like leucine provide confidence in the specificity of this assay and are described in detail previously (Fig. 3m)[33]. Upon acute virus infection, there is an increase in the frequency of kynurenine+ NK cells and also an increased magnitude of kynurenine uptake in bone marrow NK cells, when compared to naive mice (Fig. 3n). We also analysed the subset of NK cells that take up kynurenine and we were able to show that mainly the double-positive (CD27+CD11b+) NK cell subset it taking up kynurenine (Supplementary Fig. 1b). The analysis of the heavy chain subunit CD98 reveals a high expression level of CD98 in bone marrow NK cells and an upregulation of CD98 in splenic NK cells after FV infection (Fig. 3o). Taken together, NK cells increase their ATP production correlating with dramatic changes in glycolytic and mitochondrial activity and demonstrate the increased activity of the L-amino acid transporters in NK cells from FV-infected mice.

**Increased NK cell metabolism following acute MCMV infection.** To understand whether metabolic changes were a general feature of NK cells responding to acute viral infection, experiments were carried out using mice infected with a virus from a different virus family, the murine cytomegalovirus (MCMV, herpesvirus). As expected, in MCMV-infected mice, splenic NK cells produced significant amounts of IFNγ (Fig. 4b), expressed high levels of granzyme B (Fig. 4c) and the activation marker CD69 (Fig. 4a). Similarly, with mice infected with FV, NK cells responding to acute MCMV infection had increased forward scatter, a measure of cell size (Fig. 4a, 2b). Likewise, with FV infection, NK cells also increase their CD71 (Fig. 4d) and CD98 (Fig. 4e) expression after MCMV infection. Transcriptomic analysis of nutrient transporters clearly showed increased numbers of several important nutrient transporter transcripts after

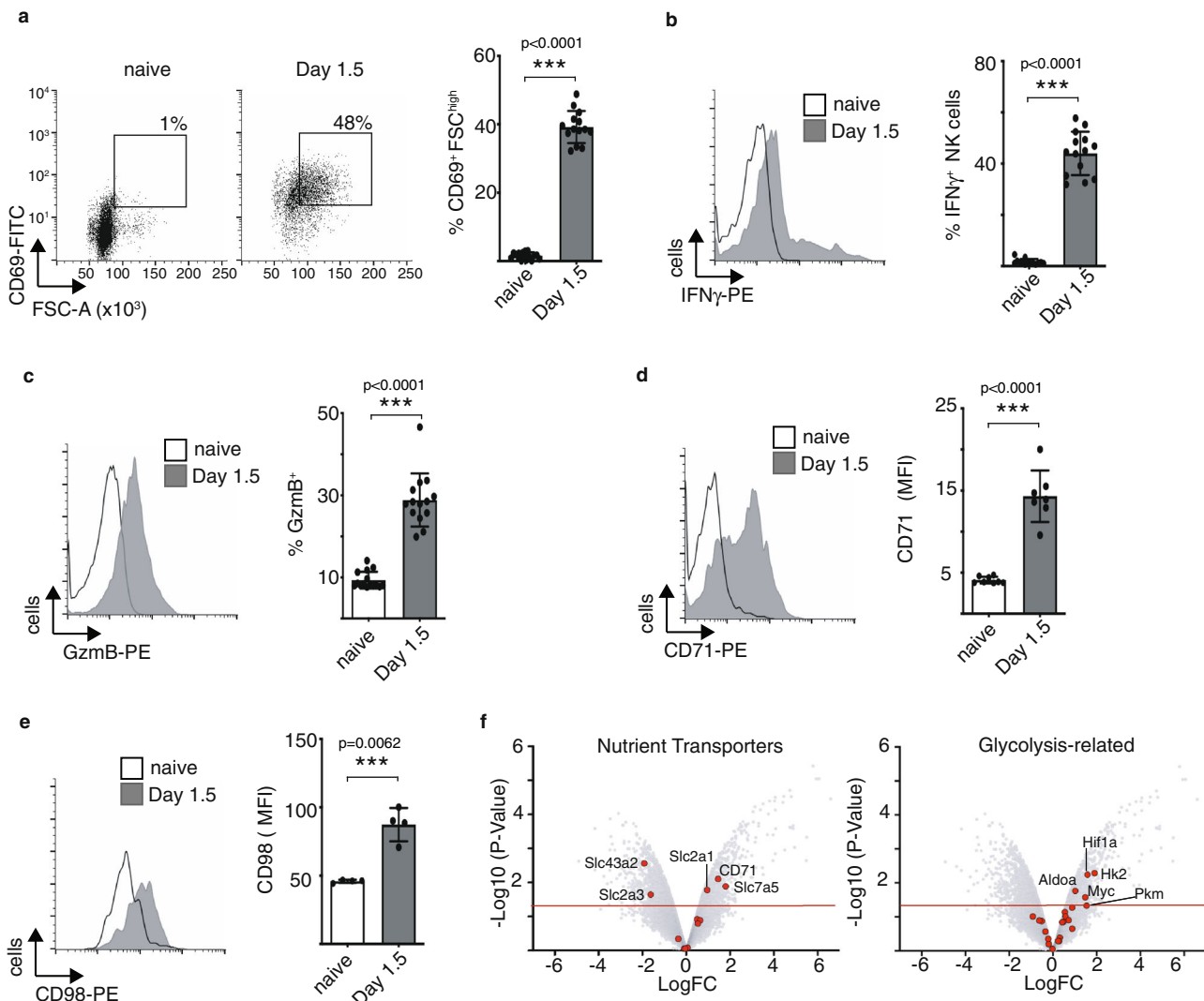

**Fig. 4 Metabolism of NK cells after acute MCMV infection.** C57BL/6 mice were infected with MCMV (Smith strain) or used uninfected and spleens were harvested at 1.5 days. NK cells (CD3−NKp46+) were analysed for activation (CD69) and cell size (FSC-A) (**a**). Ex vivo restimulated NK cells were analysed for IFNγ (**b**). Cytotoxicity was detected by granzyme B expression (**c**). A representative histogram and bar graphs of CD71 and CD98 are shown in **d** and **e**. A minimum of four (**e**) or seven (**a–d**) mice per group were used and analysed by a two-sided unpaired $t$-test and displayed ± SEM. **$p < 0.01$, ***$p < 0.001$. **f** Volcano plots of nutrient transporters and glycolysis-related genes displaying differences in relative gene expression after MCMV infection. The plot represents the −log10 of the $p$-value against log2 fold change. The horizontal red line represents a $p$-value <0.05. Source data are provided as a Source Data file.

MCMV infection, including the amino acid transporter Slc7a5, the glucose transporter Slc2a1, and the transferrin receptor CD71 (Fig. 4f). The transcriptional signatures also showed evidence of glycolytic reprogramming in NK cells after MCMV infection with a significant increase in the expression of glycolytic enzymes hexokinase 2 (Hk2), aldolase (Aldoa) and pyruvate kinase (Pkm) and glycolytic regulators Hif1α and Myc (Fig. 4f). Taken together, these data demonstrate that acute infection with the herpesvirus MCMV increases the metabolic parameters in NK cells similarly to the retrovirus infection (FV).

**Decreased cytotoxicity in Slc7a5-knockout NK cells.** It has been shown that the loss of the Slc7a5 transporter in T cells resulted in dysfunctional T cells, which could not metabolically reprogramme and did not differentiate and expand[34]. The previous data studying NK cells ex vivo have shown that the Slc7a5 transporter is important for NK cell signalling, metabolic activity and effector function[7]. To further explore the role of Slc7a5 in NK

cells responding to FV infection, we used mice with loxP sites flanking exon 2 of the Slc7a5 gene crossed with transgenic mice expressing cre recombinase under the control of the NK cell-specific NCR1 promoter (hereafter called Slc7a5NK-KO)[34,35]. First, we examined NK cells from Slc7a5NK-KO mice and compared absolute NK cell numbers in spleen, bone marrow, blood and lymph nodes with NK cell numbers from Slc7a5NK-WT mice (Fig. 5a). We found similar absolute NK cell numbers from Slc7a5NK-WT and Slc7a5NK-KO mice in all organs analysed. Next, we determined the maturation status of NK cells by measuring CD27 and CD11b on NK cells (Fig. 5b). While there was a subtle shift in the balance between CD27+CD11b− and CD27+CD11b+ NK cell subsets in the bone marrow, there were no significant differences in the differentiation states of Slc7a5NK-WT and Slc7a5NK-KO NK cells in peripheral tissues such as the spleen (Fig. 5b). Then, we validated that cytokine-stimulated NK cells from the Slc7a5NK-KO mice were deficient for Slc7a5 activity by measuring kynurenine uptake into splenic NK cells (Fig. 5c). NK

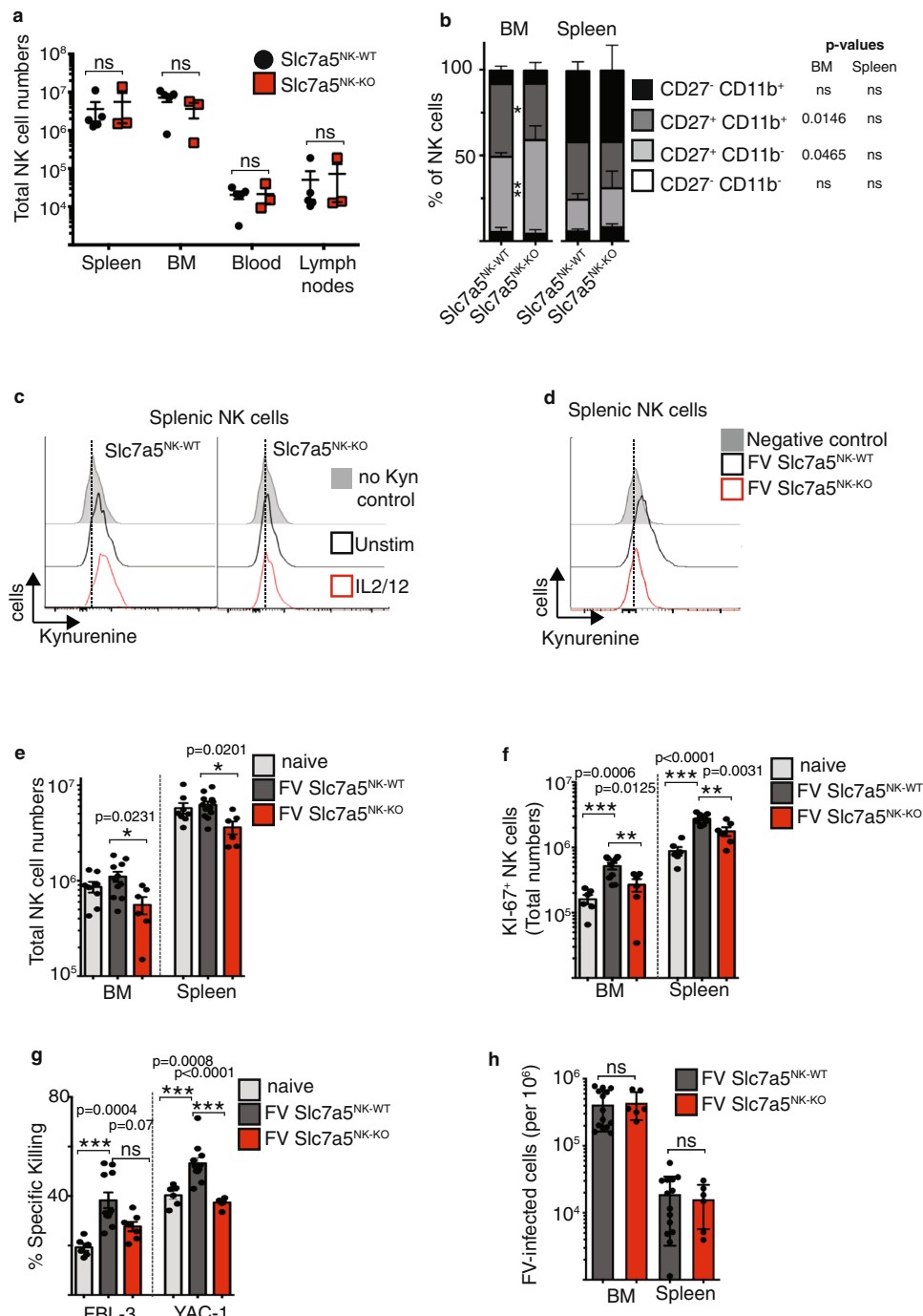

**Fig. 5 Influence of Slc7a5 deletion in NK cells during acute retrovirus infection.** Spleen, bone marrow (BM, tibia and femur of the right hind leg), lymph nodes (axillary, brachial, inguinal) and blood were collected from mice with loxP sites flanking exon 2 of the Slc7a5 gene crossed with transgenic mice expressing cre recombinase under the control of the NCR1 promoter (Slc7a5NK-WT and Slc7a5NK-KO mice). Single-cell suspensions were counted, stained for NK cell markers (CD3−CD19−NK1.1+NKp46+) and measured by flow cytometry (**a**). Cell numbers in the blood were calculated per ml of blood. A minimum of three mice from two independent experiments were used and analysed by Mann–Whitney test. Data are presented as mean values ± SEM. Frequencies of CD27 and CD11b NK cell subsets of Slc7a5NK-WT and Slc7a5NK-KO mice are displayed in **b** for bone marrow and spleen. Data are presented as mean values ± SEM. NK cell kynurenine (kyn) uptake of unstimulated or IL-2/12-stimulated NK cells is displayed in **c**. Slc7a5NK-WT and Slc7a5NK-KO were infected with FV and at 7 dpi, the spleen and bone marrow of FV-infected Slc7a5NK-WT, Slc7a5NK-KO, or naive mice were harvested. A representative histogram from splenic NK cells (CD3−CD49b+ NK1.1+) from Slc7a5NK-WT and Slc7a5NK-KO mice is shown in **d**. Experiments were repeated independently twice with similar results. The absolute cell numbers of NK cells (**e**) and their proliferation (**f**) are displayed as bar graphs ± SEM. YAC-1 cells (right-hand side) or FV-induced tumour cells (FBL-3 cells, left-hand side) were stained with Tag-it-violet and co-incubated with isolated, splenic NK cells for 18 h (**g**). Viral loads were analysed by an IC assay in Slc7a5NK-WT and Slc7a5NK-KO mice (**h**). Data are presented as mean values ± SEM (**g**, **h**). A minimum of six mice per group from two independent experiments was used (**e–h**) and analysed by ordinary one-way ANOVA (**b**, **e**, **f**), unpaired *t*-test (**h**) or Kruskal–Wallis test (**g**). Significances are indicated as follows: **p < 0.01, ***p < 0.001. Source data are provided as a Source Data file. BCH 2-Amino-2-norbornanecarboxylic acid, ns not significant.

cells from Slc7a5[NK-WT] mice showed kynurenine uptake into splenic NK cells and a further increase when NK cells were stimulated with the cytokines IL-2/IL-12 overnight. In contrast, Slc7a5[NK-KO] NK cells were completely deficient for kynurenine uptake. We addressed the question of whether we can find differences in the expression of effector molecules and metabolic parameters at steady state in Slc7a5[NK-WT] and Slc7a5[NK-KO] mice (Supplementary Fig. 2a). No differences were detectable between WT and KO mice. Next, we infected knockout and control mice with FV and analysed the kynurenine uptake after infection (Fig. 5d). Slc7a5[NK-WT] NK cells were able to take up kynurenine whereas Slc7a5[NK-KO] mice lost the ability to take up kynurenine and showed similar levels as compared to the negative control BCH (System L blocker—no uptake of kynurenine). Additionally, we analysed the kynurenine uptake of T cells in FV-infected mice. T cells in Slc7a5[NK-WT] and Slc7a5[NK-KO] mice transported kynurenine and no differences were observed, thus verifying the NK cell specificity of the knockout (Supplementary Fig. 2b). Next, we analysed the absolute numbers of NK cells in Slc7a5[NK-WT] and Slc7a5[NK-KO] mice after FV infection (Fig. 5e). We detected significantly lower splenic and bone marrow NK cell numbers in Slc7a5[NK-KO] mice compared to Slc7a5[NK-WT] mice. Correlating with these data, we detected more proliferating KI-67[+] NK cells (absolute numbers) in Slc7a5[NK-WT] mice in comparison to Slc7a5[NK-KO] mice in both analysed organs after retrovirus infection (Fig. 5f). However, the frequency of KI-67 positive NK cells from Slc7a5[NK-WT] and Slc7a5[NK-KO] mice was similar after FV infection (Supplementary Fig. 2c). Interestingly, NK cells from Slc7a5[NK-WT] and Slc7a5[NK-KO] mice exhibit comparable levels of activation and cytokine production, shown by CD69, TNF and IFNγ expression (Supplementary Fig. 2e). No differences in viability were detectable after FV infection in Slc7a5[NK-WT] and Slc7a5[NK-KO] mice (Supplementary Fig. 2d). Additionally, the overall IFNγ concentration in the serum of Slc7a5[NK-WT] and Slc7a5[NK-KO] mice were similar (Supplementary Fig. 2f). NK cell metabolic parameters such as the NK cell size, CD71, transferrin uptake, mitochondrial mass and polarisation and cMyc expression were also not significantly affected by the absence of Slc7a5 activity (Supplementary Fig. 2g). Next, we performed an in vitro cytotoxicity assay and co-incubated isolated NK cells from infected Slc7a5[NK-WT] and Slc7a5[NK-KO] mice with tumour cell targets (YAC-1, Fig. 5g) and FV-derived tumour cells (FBL-3, Fig. 5g). As expected, NK cells from FV-infected Slc7a5[NK-WT] mice had an increased killing capacity compared to cells from naive animals. However, NK cells from Slc7a5[NK-KO] mice showed significantly reduced levels of target cell killing compared to Slc7a5[NK-WT] cells. Next, we analysed the viral loads of FV-infected Slc7a5[NK-WT] and Slc7a5[NK-KO] mice (Fig. 5h). Although we detected the differences in target cell killing, there were no differences in the viral burden in infected Slc7a5[NK-WT] and Slc7a5[NK-KO] mice in both analysed organs. Collectively, these data show that the Slc7a5 transporter has an influence on NK cell numbers and cytotoxicity against YAC-1 target cells during viral infection but is not required for the overall NK cell antiviral response.

**Iron availability is important for NK cell activity during FV infection.** A well-characterised cMyc target gene is the transferrin receptor CD71, which is an important nutrient uptake receptor that mediates iron uptake via the endocytosis of transferrin-bound iron into the cell[10]. Iron is essential for cellular biochemistry, and T cell mitochondrial metabolism and effector functions are dependent on iron uptake[12]. Considering that CD71 expression was increased on NK cells in mice infected with MCMV (Fig. 4d), we hypothesised that iron uptake could be

important for the NK cell response against retroviral infection. To analyse the uptake of iron into NK cells, we measured fluorescently labelled transferrin as an indicator of iron uptake by flow cytometry[11]. There was an increase in the uptake of transferrin into NK cells in FV-infected mice compared to NK cells from naive mice (Fig. 6a, b). Competition with unlabelled transferrin and performing the assay at 4 °C are important controls that provide confirmation in the specificity of this assay (Fig. 6a)[11]. Increased transferrin uptake and CD71 expression were observed in both NK cell populations from the bone marrow and the spleen following infection (Fig. 6b, c), suggesting that responding NK cells have an increased demand for iron. We showed that CD71 expression and transferrin uptake were mainly present by CD27[+]CD11b[+] NK cells (Supplementary Fig. 1b). Viruses such as HIV alter iron levels and recent publications show that infection with SARS-CoV-2 result in hypoferremia[36,37] but it remains unclear how these infection-associated shifts in iron availability might alter the outcome of viral infection[38]. Here we addressed the question of whether serum iron levels influence antiviral NK cell activity. Hepcidin is a hormone that prevents the export of iron into the serum by binding to the iron export channel protein ferroportin[39]. Elevated levels of hepcidin result in a reduction of serum iron. We used a synthesised polypeptide called mini-hepcidin that acts analogously to hepcidin to reduce iron levels in mice[12,40]. We injected mini-hepcidin in the vehicle DSPE or DSPE alone every day starting at the day of FV infection (Fig. 6d). The measurement of serum iron showed a more than ten-fold decrease of serum iron in mice receiving mini-hepcidin (Fig. 6e). Additionally, we detected increased CD71 expression on NK cells from infected, mini-hepcidin-treated mice in comparison to vehicle-treated, infected control mice indicating relative cellular iron deficiency (Fig. 6f), which is consistent with hypoferremia (Fig. 6e). Interestingly, the analysis of NK cell activation revealed a significant decrease of CD69[+] NK cells in animals with reduced levels of serum iron upon acute FV infection in the spleen and bone marrow (Fig. 6g). Similarly, the percentage of IFNγ[+] NK cells was decreased after mini-hepcidin treatment (Fig. 6h). The measurement of the overall IFNγ concentration in the serum of mice revealed a significant increase of IFNγ concentration in the serum of FV-infected mice compared to naive mice (Fig. 6i). After mini-hepcidin (mHep) treatment the IFNγ levels were significantly reduced compared with FV-infected mice and comparable with serum from naive mice. Due to the systemic inhibition of iron export into the serum using mHep, we addressed the question of whether NK cell functions are altered due to different levels of the cytokines responsible for NK cell activation in response to FV, including IL-15 and IL-18[4]. In Fig. 6j, we were able to show that the cytokine concentrations were not decreased upon mHep treatment but in fact, IL-12, IL-15 and IL-18 levels were significantly increased. In line with Fig. 6h, i, we detected reduced mRNA levels of IFNγ and the IFNγ-induced chemokine CXCL10 in splenocytes (Fig. 6k), but we did not detect differences in the levels of the pro-inflammatory cytokines IL-2, IL-15 and IL-18, previously described to be important for NK cell activation upon FV infection and predominantly secreted by macrophages, DCs or CD4[+] T cells[4,18]. Together, these data show that NK cells have an increased demand for iron when responding to acute FV infection and that experimental reduction of serum iron levels inhibits NK cell activation and their IFNγ production.

**Iron deprivation inhibits NK cell responses against FV infection.** We observed dramatic effects of iron deprivation on NK cell activation and cytokine production. We next assessed whether low iron concentrations have an influence on metabolic

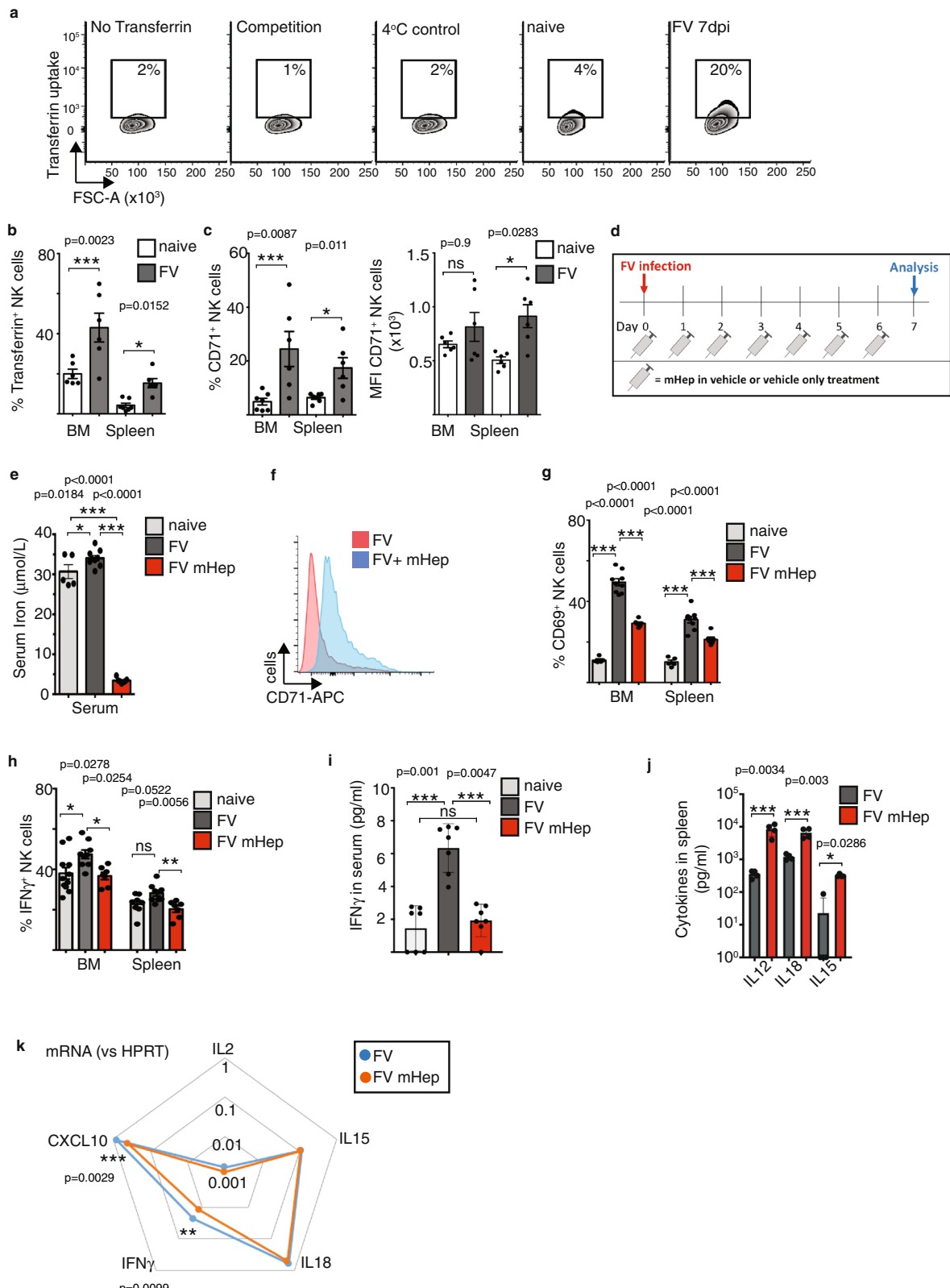

parameters of NK cells such as cell growth and cMyc expression. Thus, we analysed the NK cell forward scatter as a measure of NK cell size in low serum iron groups and control mice. We detected a smaller NK cell size in the FV-infected, iron-deprived group compared to FV-infected, vehicle-treated mice (Fig. 7a). The small cell size of these NK cells was comparable to that of NK cells from naive, vehicle-treated mice and was observed in both

the spleen and bone marrow. We also discovered a dramatic decrease of cMyc+ NK cells in mice with low serum iron upon FV infection (Fig. 7b). To further explore the influence of iron deficiency on the NK cell capacity to kill target cells, we performed a cytotoxicity assay using FV-induced tumour cell targets (FBL-3 cells, Fig. 7c). NK cells isolated from infected mice with low serum iron levels showed significantly less killing of FBL-3

**Fig. 6 Iron uptake by NK cells upon FV infection.** C57BL/6 mice were infected i.v. with FV for 7 days or used as uninfected, naive control. Transferrin uptake assay and controls (no transferrin, competition with holo-transferrin, uptake at 4 °C) are shown as representative histograms for splenic NK cells (**a**). Experiments were repeated independently twice with similar results. Transferrin uptake of NK cells (**b**) and CD71$^+$ NK cells (**c**) from spleen and bone marrow (BM) are displayed as bar graphs ± SEM. Statistically significant differences were analysed by two-tailed Mann–Whitney test (**b**) and two-tailed unpaired $t$-test (**c**). Minimum of six mice per group from two independent experiments was used for the analysis. An experimental overview of the induction of low serum iron concentrations using mini-hepcidin (mHep) is shown in box **d**. Serum iron was analysed in **e** with a minimum of five mice from two independent experiments. Data are presented as mean values ± SEM. A representative histogram of CD71 expression on NK cells of FV-infected vehicle-treated and FV-infected mHep-treated mice is shown in **f**. Experiments were repeated independently twice with similar results. Activation of NK cells was analysed by the early activation marker CD69 (**g**) and the percentage of IFNγ$^+$ NK cells (**h**) after mHep treatment are displayed as bar graphs ± SEM. At least six mice from two independent experiments were used. IFNγ concentration in mouse serum (**i**) as well as IL-12, IL-15 and IL-18 concentrations in spleens (**j**) were analysed using a custom-made Legendplex kit. Cytokines were measured in a minimum of seven mice from two independent experiments (**i**) or four mice from one experiment (**j**) were used for the analyses. In **i** serum from seven mice from two independent experiments were used. At least four mice were used to measure cytokines in the spleen from one experiment (**j**). Data are presented as mean values ± SEM (**i**, **j**). mRNA levels of cytokines were measured in splenocytes by quantitative real-time PCR and are displayed as spider plot (**k**). Statistically significant differences were analysed by an ordinary one-way ANOVA within groups of bone marrow or spleen. Significances are indicated as follows: *$p < 0.05$, **$p < 0.01$, ***$p < 0.001$. Source data are provided as a Source Data file. ns not significant.

target cells than NK cells isolated from infected mice with normal serum iron levels. These NK cells eliminated approximately twice as many target cells as the iron-deprived NK cells. With the knowledge that low serum iron results in low activation, cytotoxicity and impaired metabolic reprogramming of NK cells, we then assessed the influence of low iron levels on viral loads of FV-infected mice (Fig. 7d). We observed significantly increased viral loads in the bone marrow, the main reservoir during initial viral infection, of mice treated with mini-hepcidin in comparison to vehicle-treated control mice. Considering that the levels of cytokines responsible for NK cells activating in FV-infected mice were not reduced in mice with low serum iron, we investigated whether limited iron availability has a direct effect on NK cells. Thus, we took advantage of in vitro expanded NK cells using an increasing concentration of an iron chelator deferoxamine (DFO) to decrease the iron content in the medium (Fig. 7e–h). To confirm that any observed effects of DFO were due to reduce iron availability a control was included with the highest dose of DFO supplemented with exogenous iron in the form of FeSO$_4$. NK cells were stimulated with cytokines known to promote metabolic function and NK cell activation and there was a significant dose-dependent decrease in NK cell size (Fig. 7e), mitochondrial mass (Fig. 7f) as well as perforin (Fig. 7g) and granzyme B (Fig. 7h) in cultures with reduced iron levels. Importantly, adding back iron (FeSO$_4$) in iron-deprived NK cell cultures restored these metabolic and functional outputs of NK cells. It has been shown previously that NK cell depletion and subsequent FV infection leads to increased viral loads, thus demonstrating the importance of NK cells in controlling this retrovirus[41]. Given the data showing that NK cells have impaired effector functions and metabolism in hypoferremic, FV-infected mice, we hypothesised that depleting NK cells from these mice would not affect viral loads because dysfunctional NK cells were not able to control FV infection. Therefore, we infected mice with FV and induced hypoferremia by repetitive injections of mHep. Additionally, one group of mice were depleted for NK cells. More than 92% of NK cells were depleted after the injections of NK cell depletion antibody NK1.1 (Fig. 7i). The analysis of the viral loads revealed no differences between the mHep-treated and mHep, NK cell-depleted groups (Fig. 7j). These data collectively show that sufficient serum iron is essential for NK cell metabolic responses and antiviral activity, and that serum iron deficiency can undermine the control of acute retroviral infection.

## Discussion

Viruses are the cause of numerous human diseases worldwide. The urgent need in understanding viral mechanisms and host responses as well as potential antiviral treatments are not only fundamental during the ongoing pandemic due to the new viral Coronavirus disease 2019 (COVID-19), but also for other viral infections that still cause health issues without vaccinations or cure (e.g. Dengue, Herpes simplex, HIV). Cellular metabolism is a very important parameter for multiple aspects of viral infection. Viruses such as HIV reprogram the cells that they infect towards increased levels of cellular metabolism, most notably increased glycolysis, as this increases the biosynthetic capacity of the cells to generated new viral particles[42]. As the study of immune metabolism reveals novel therapeutic strategies across all immunology fields, we wanted to understand the importance of metabolic processes for the antiviral functions of NK cells. As part of the innate immune system, NK cells respond rapidly and in vitro studies demonstrate that changes in metabolism are not required for the rapid (3–4 h) production of cytokines such as IFNγ and for NK cell-mediated cytotoxic killing of target cells[43]. However, NK cells can also operate over longer periods during antiviral responses and NK cells stimulated with cytokines for 24 h undergo clear metabolic changes that are essential for effector functions. Therefore, from the outset of this study, it was extremely uncertain whether cellular metabolism would be important for NK cells responding to acute FV infection in vivo during the innate phase of this antiviral response.

In this study, we demonstrate for the first time that there is increased metabolic reprogramming and nutrient uptake by NK cells responding to acute viral infections in vivo, in this case to acute Friend retrovirus (FV) infection and murine cytomegalovirus (MCMV). The data show that there is a distinct and robust metabolic response at the peak of the NK cell response to FV, at 7 dpi, that involved increased nutrient uptake and elevated rates of glycolysis and OXPHOS. NK cell glycolysis has previously been implicated in the response to MCMV infection as mice treated with 2-deoxyglucose (2DG), an inhibitor of glucose metabolism, showed increased MCMV viral burden[44]. However, it is worth noting that 2DG is not a specific inhibitor of glycolysis as 2DG inhibits other glucose-dependent processes including protein glycosylation. In FV-infected mice, NK cells show increased levels of cMyc protein, a metabolic regulator that has been shown to be important for mediating increased glycolysis and OXPHOS in ex vivo activated NK cells[7]. cMyc also regulates the capacity of cells to uptake key nutrients including amino acids and iron. The transferrin receptor CD71 is a well-characterised cMyc target gene, as are amino acid transporters including Slc1a5 and Slc7a5, important for the uptake of glutamine and large neutral amino acids, respectively[10,45–47]. Indeed, we show that in parallel with elevated cMyc expression, NK cells

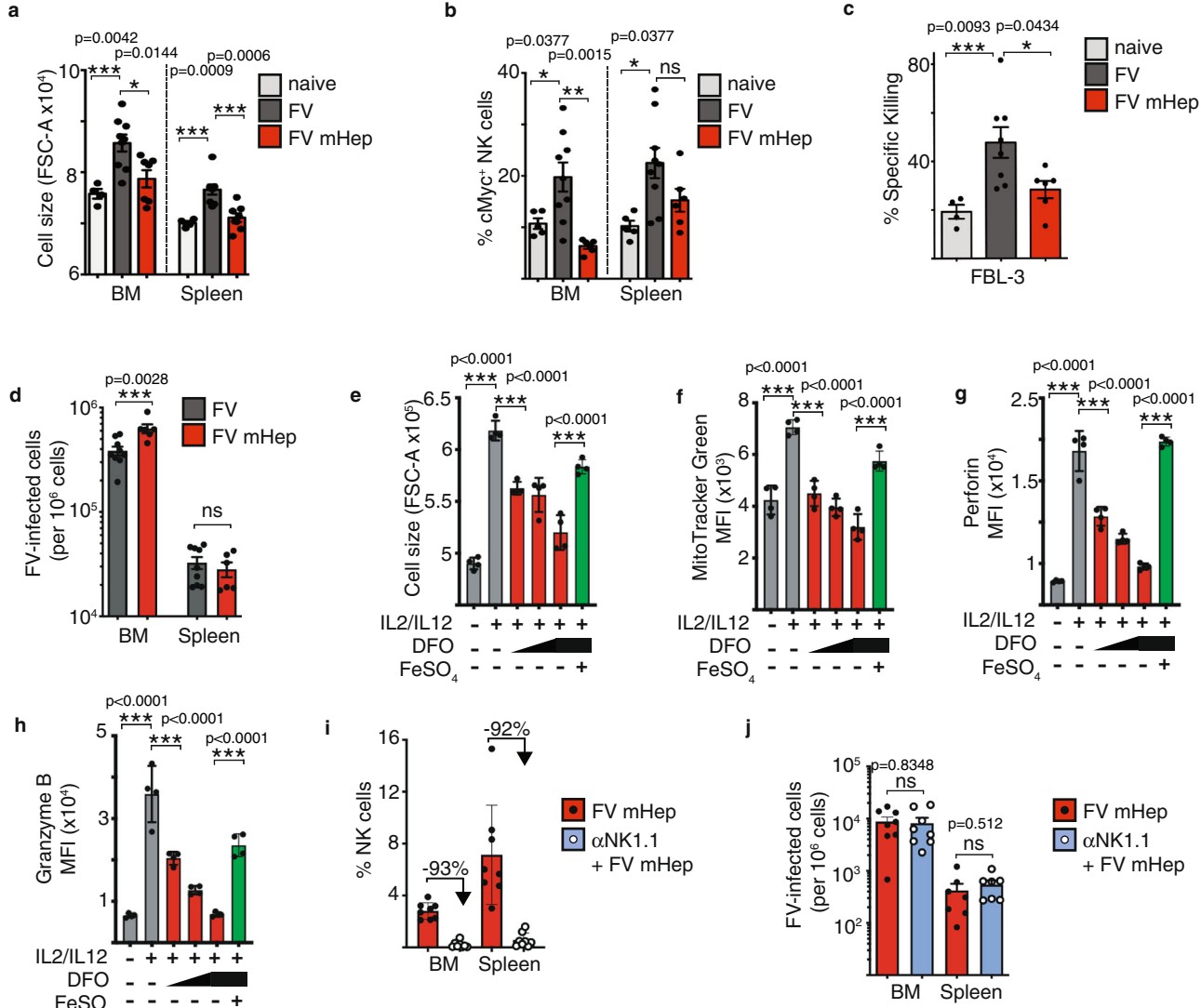

**Fig. 7 Influence of low serum iron on NK cells during acute FV infection.** C57BL/6 mice were infected i.v. with FV for 7 days or used as uninfected, naive control. Mice were treated every day i.p. with DSPE (vehicle) or mini-hepcidin (mHep) in DSPE. At the day of the experiment, bone marrow (BM) and spleens were harvested and stained for analysis by flow cytometry. NK cells were analysed for cell size (FSC, **a**) and for cMyc (**b**). NK cells were isolated from spleens of naive + vehicle and FV-infected mice (FV + vehicle and FV + mHep) and co-incubated with Tag-it-Violet stained FV-induced FBL-3 tumour cells. After co-incubation, co-culture was stained for viability and analysed by flow cytometry (**c**). Viral loads were detected by an Infectious Center assay in bone marrow and spleen and analysed by a two-sided Mann–Whitney test (**d**). At least five mice per group from two independent experiments were used. Statistically significant differences between the groups were analysed with an Ordinary one-way ANOVA (**a**–**c**). The cell culture medium of in vitro expanded NK cells was modified for iron availability (DFO) and additional iron (FeSO$_4$) was added if indicated. NK cells were analysed for cell size (**e**), MitoTracker Green (**f**), perforin (**g**) and granzyme B (**h**). Experiments were performed once with four biological replicates. Statistically significant differences were analysed by an ordinary one-way ANOVA with a Sidak post-test. NK cells (NK1.1$^+$CD4$^-$CD8$^-$) were depleted with an anti-NK1.1 depletion antibody and depletion efficiency is shown in **i**. At 3 dpi, viral loads of FV-infected, mHep-treated ± NK cell depletion is displayed in **j** and were analysed by a two-sided unpaired $t$-test. All data are presented as mean values ± SEM. $*p < 0.05$, $**p < 0.01$, $***p < 0.001$. Source data are provided as a Source Data file. ns not significant.

in FV-infected mice had increased expression of CD71 and associated transferrin uptake, as well as increased amino acid uptake through Slc7a5. It was previously shown that cMyc is required in NK cells for augmented glucose transporter expression and glycolytic enzymes as well as to support mitogenesis leading to increased mitochondrial mass[7]. During acute FV infection, cMyc expression in NK cells is associated with increased mitochondrial mass and membrane potential and elevated mitochondrial respiration. Taken together, it is clear that acute FV infection drives metabolic changes in NK cell characteristics of cMyc-mediated metabolic regulation.

Having characterised substantive metabolic changes in NK cells responding to FV, the functional importance of this NK cell metabolic phenotype was then investigated. Pharmacological inhibition of Slc7a5-mediated amino acid uptake has previously been shown to disrupt the function of NK cells stimulated with cytokines ex vivo[7]. A previous study by Sinclair et al. analysed the importance of Slc7a5 in T cells using a T cell-specific Slc7a5-knockout mouse[34]. They showed impaired T cell clonal expansion and effector differentiation as well as dysfunction in the metabolic reprogramming in Slc7a5-null T cells[34]. Here we have generated mice in which Slc7a5 is depleted specifically in

NK cells. NK cells develop normally in these mice but show discrete defects following FV infection. In Slc7a5[NK-KO] mice infected with FV, we detected a significant decrease in NK cell numbers, which is not observed in Slc7a5[NK-KO] mice without infection. Interestingly, the NK cells show similar frequencies of KI-67 staining in Slc7a5[NK-KO] and Slc7a5[NK-WT] mice at 7 dpi, which marks cells that are engaging in cell cycling. In addition, the maturation stages of Slc7a5-null NK cells were similar to the Slc7a5-WT NK cells after FV infection (data not shown). Therefore, the reduced numbers of NK cells observed in FV-infected Slc7a5[NK-KO] mice may be due to reduced rates of cell cycle progression or alternatively to alterations in NK cell trafficking. Another possibility is that the NK cell response is delayed due to metabolic adaptation to the use of other amino acid transporters. This idea is supported by the observation that cMyc levels are not affected in Slc7a5-null NK cells in FV-infected mice. While cMyc can promote the expression of Slc7a5, it has also been shown in NK cells and T cells that amino acid uptake through Slc7a5 is important for the protein expression of cMyc[7,34]. In NK cells stimulated with cytokines ex vivo, blocking Slc7a5 activity results in the rapid loss of cMyc protein expression[7]. Therefore, cMyc and Slc7a5 activities are closely interconnected[46]. In Slc7a5-null NK cells from 7 days FV-infected mice, cytokine production, as well as metabolic factors such as cell size, transferrin uptake and cMyc, were similar to FV-infected Slc7a5[NK WT] mice despite the loss of amino acid uptake through Slc7a5. These data suggest that NK cells have adapted to uptake amino acids through alternative transporters to sustain cMyc protein expression or alternatively have adapted to using an alternative mechanism for cMyc regulation that is independent of amino acid availability. In contrast, Slc7a5-deficiency has a striking effect on the in vivo functions of T cells including cytokine production and granzyme B expression[45]. Taken together, these data suggest that NK cells possess a capacity for metabolic adaptation that T cells may lack. This idea is supported by a recent study that demonstrated NK cells have the capacity for metabolic adaptation in the absence of a key glycolytic enzyme, pyruvate kinase muscle isoform 2 (PKM2)[48]. Despite clear metabolic and functional defects in NK cells when PKM2 activity was manipulated in ex vivo cultured NK cells, mice lacking PKM2 specifically in NK cells showed completely normal immune responses to MCMV infection. However, while NK cells in FV-infected mice have adapted to maintain cMyc expression and cytokine production in the absence of Slc7a5, there are some clear functional consequences of Slc7a5 deletion as we observed lower cytotoxicity in Slc7a5-null NK cells during acute retroviral infection.

Another role of cMyc is the regulation and fine-tuning of transferrin receptor expression on immune cells, thus cMyc is important in controlling iron metabolism[10,11]. Iron is crucial for cellular processes such as DNA synthesis, DNA repair and multiple metabolic processes including OXPHOS. Systemic iron is tightly regulated by the hormone hepcidin. Dysregulation of iron homoeostasis can cause several diseases and viruses can also affect iron levels, which are linked to the severity of infections[49]. Recently it was reported that COVID-19 is associated with hypoferremia[37] and viruses like hepatitis B, hepatitis C and HIV also alter iron homoeostasis[36,49]. After FV infection, NK cells upregulate the transferrin receptor and take up increased amounts of transferrin-bound iron. In this study, serum iron concentrations were suppressed by repetitive injections of mini-hepcidin and we analysed the impact of reduced serum iron on antiviral NK cell responses. Herein, we demonstrate that NK cells are very sensitive to serum iron deprivation, showing increased CD71 expression consistent with cellular iron deficiency. In contrast, macrophage- and DC-derived cytokines, such as IL-15

and IL-18, which are important in NK cell activation in FV-infected mice, were not decreased suggesting a direct effect of iron on NK cells. The data showed that in iron deficiency, NK cells have defects in their activation and metabolism, as the NK cells were small and had reduced cMyc levels that were comparable to those of NK cells from uninfected, control-treated mice. Similarly, when iron levels were reduced using the iron chelator deferoxamine during in vitro activation NK cells showed evidence of impaired metabolism and function, providing further evidence for a direct role of iron in facilitating NK cell responses. Interestingly, in iron-deficient mice infected with FV, we were able to show a dramatic decrease in NK cell cytotoxicity and observed an associated increase in bone marrow viral load, the organ of NK cell origin and high viral replication. While NK cells deficient in the expression of the amino acid transporter Slc7a5 showed a degree of metabolic plasticity that allowed NK cells to sustain much of their antiviral function, NK cells were not able to adapt to the low iron levels in mini-hepcidin treated mice. This likely reflects the fact that iron is a crucial biological metal required as a cofactor for numerous enzymes, most notable the complexes of the mitochondrial electron transport chain. Indeed, one of the clearest consequences of depleting iron with deferoxamine during in vitro NK cell activation was the inability of these NK cells to increase cellular mitochondrial mass.

These findings have important therapeutic implications, suggesting that iron supplementation may improve antiviral NK cell responses, especially in iron-deficient patients. It might also be interesting to determine whether NK cell metabolic activity and antiviral functions can be further augmented in iron-sufficient settings by iron supplementation or whether raised iron would have negative effects such as renal toxicity, cardiovascular and immunologic reactions or viral rebound[50–52]. However, the clinical use of iron preparations has increased in recent years and is worth exploring in the context of viral infection[53].

Viruses such as CMV, hepatitis C and HIV require the host cell to facilitate the biosynthesis of new viral particles. The host cell iron level is important in the process[54]. These viruses remodel the iron uptake and/or processing within the host cells using various viral proteins[55,56]. In HIV infection accumulation of iron, mostly in macrophages, was shown to promote the HIV replication and is associated with poor outcome in HIV-1 infection[57]. Using iron chelators in order to lower the iron availability to infected cells suppressed viral replication using in vitro approaches[54]. For this reason, it has been proposed that lowering iron levels to disrupt the life cycle of the virus could be a beneficial therapeutic approach[38]. In our study, we make the novel discovery that serum iron deficiency has dramatic detrimental effects on the NK cell antiviral immune response and results in increased viral burden in vivo. Overall, this indicates that depriving immune cells of iron and limiting their antiviral functions has a greater negative impact on viral burden than any positive effects iron deficiency has through slowing viral replication in virus-infected cells. This is an important and novel finding that argues that reduced serum iron may contribute to poor outcomes for patients infected with retrovirus, at least in part due to disrupting NK cell responses. One limitation of this is model is that mini-hepcidin injection does not selectively deprive NK cells of iron and instead causes physiological serum iron deficiency which recapitulates the hypoferremia observed during some human infections. Therefore it will be interesting to examine to which extent experimentally induced iron deficiency perturbs the acute response to retroviral infections through direct effect on NK cells or via other immune cells. Interestingly we do not detect differences in the expression level of cytokines that have been shown previously to be highly important for supporting the activity of NK cells during acute FV infection[4,18].

mRNA levels for IL-15 and IL-18, produced by macrophages and dendritic cells and IL-2, secreted by activated CD4+ T cells, were equivalent in iron-deficient and iron-sufficient mice infected with FV. Independent of a direct or indirect effect, iron has a strong effect on the metabolic phenotype and the cytotoxicity of NK cells.

The currently used antiretroviral therapy (ART) for treating patients with HIV diminishes the viremia below the detection limit, but the eradication of HIV is still not possible. HIV-1 persists in cellular reservoirs and upon treatment interruption rapid viral rebound occurs[58]. During HIV infection, but also other persistent viral infections such as infections with FV, Lymphocytic choriomeningitis virus (LCMV), hepatitis B and hepatitis C, both cytotoxic CD8+ T cells and NK cells become dysfunctional and lose their cytotoxic potential[59–62]. Interestingly, CD8+ T cell and NK cell responses during persistent HIV infection can be reinvigorated with the pro-inflammatory cytokine IL-15, which had a beneficial effect on survival, memory formation and metabolic plasticity[21,63–65]. In this study, persistent HIV-infected individuals co-infected with HCMV have slight mitochondrial defects and less fuel flexibility in their adaptive NK cell population[21]. Herein we show that during acute retrovirus infection NK cells enhance their OXPHOS as well as their mitochondrial activity, resulting in increased ATP production. This increased energy generation is necessary to sustain the augmented proliferation and cytokine production of NK cells during acute infection. It seems that maintaining mitochondrial fitness is important for sustaining the antiviral activity of NK cells. Indeed, the induction of mitophagy, a process essential for clearing dysfunctional mitochondria has been shown to be crucial in NK cells response to MCMV infection in mice[66]. Therefore, it is tempting to speculate that targeting NK cell mitochondrial fitness may be an effective new therapeutic approach for retroviral infections.

## Methods

**Animals**. C57BL/6, Slc7a5fl/fl (kindly provided by Linda Sinclair, University of Dundee, UK) and NCR1 Cre mice (kindly provided by Veronika Sexl, the University of Veterinary Medicine of Vienna, Austria) were bred in house. Male and female mice used in this study were on a C57BL/6 background and between 6 and 12 weeks of age. Mice were housed under 12:12 light cycle in a relative humidity of 45–65% and a temperature between 20 and 24 °C. Mouse experiments were approved and in compliance with the Irish Health Products Regulatory Authority and the Animal Research Ethics Committee (AREC) at Trinity College Dublin.

**FV infection and Infectious Center assay**. At least six-weeks-old mice were used for experiments. The FV complex containing B-tropic Friend murine leukaemia helper virus and polycythaemia-inducing spleen focus-forming virus was used for infections. Mice were injected intravenously with 0.1 ml phosphate-buffered saline (PBS) containing 40,000 SFFU of FV. The virus stock did not contain lactate dehydrogenase-elevating virus. Detection of infectious centres was done by 10-fold dilutions of single-cell suspensions (spleen, bone marrow) onto Mus dunni cells. For the bone marrow group, femur and tibia of hind legs were used. Cells were incubated for 72 h and fixed with EtOH. Cells were stained with the F-MuLV envelope-specific monoclonal antibody 720, and developed with a peroxidase-conjugated goat anti-mouse antibody (1:400, 115-035-003, Dianova). Finally, cells were incubated with aminoethylcarbazol (A6926-100TAB, Sigma) and washed with water for the detection of foci.

**Flow cytometry**. Surface and viability stains were done at room temperature for 20 min in the dark. Cells were fixed with Cyto-Fast Fix-Perm Buffer Set (426803, BioLegend) and stained intracellular in corresponding buffer. For intranuclear staining, cells were fixed with Foxp3/Transcription Factor Fixation/Permeabilization Kit (00-5523-00, eBioscience). Cells were restimulated with ionomycin (500 ng/ml, I9657, Sigma), phorbol myristate acetate (PMA; 100 ng/ml, P8139, Sigma), BFA (5 μg/ml, 420601, BioLegend), Monensin (2 μg/ml), 420701, BioLegend, diluted in supplemented RPMI buffer at 37 °C for 3 h. Dead cells were excluded using Zombie Aqua (1:1000, 423102, BioLegend). Cells were analysed with Canto II flow cytometer (BD) and BD DIVA 8.0 software. Antibodies used as follows: CD3 (17A2, FITC, 1:200, 100204, BioLegend), CD11b (M1/70, PE Cy7, 1:400, 101216, BioLegend), CD27 (LG.3A10, PE, 1:200, 558754, BD Pharmingen),

CD49b (DX5, APC-Vio 770, 1:200, 130-105-249, Miltenyi Biotech), CD69 (H1.2F3, PerCP-Cy5.5, 1:200, 561931, BD Pharmingen), CD71 (R17217, APC, 1:200, 17-0711-82, eBioscience), CD98 (RL388, PE, 1:200, 12-0981-81, eBioscience), cMyc (D84C12, PE, 1:100, 14819, Cell Signaling), FasL (MFL3, PerCP-eFluor 710, 1:200, 46-5911-82, eBioscience), GzmB (NGZB, PE Cy7, 1:200, 25-8898-82, eBioscience), IFNγ (XMG1.2, APC, 1:100, 554413 BD Pharmingen), KI-67 (REA183, PE-Vio770, 1:200, 130-120-419, Miltenyi Biotech), NK1.1 (PK136, BV421, 1:200, 108732, BioLegend), Ter119 (TER-119, BV510, 1:200, 116237, BioLegend), TNF (MP6-XT22, PE Cy7, 1:100, 25-7321-82, eBioscience).

Cells were resuspended in complete RPMI plus Tetramethylrhodamine, methyl ester (TMRM, 100 nM, T668, Invitrogen) and MitoTracker Green (MTG, 100 nM, M7514, Invitrogen) and incubated for 20 min at 37 °C. As a control, Oligomycin (2 μM, Sigma) and FCCP (20 μM, Sigma) was added to cells. For CellROX staining (C10491, Invitrogen), controls were incubated with Tert-butyl Hydroperoxide (TBHP, 200 μM) for 1 h at 37 °C. CellROX was added for 20 min at 37 °C (5 μM). Cells were measured without fixation. Data were analysed with FlowJo Version 10.

**Transferrin uptake**. Cells were washed with serum-free RPMI supplemented with 0.5% BSA twice. Spleen cells were incubated with or without 5 μg/ml of Alexa Fluor 647-conjugated Transferrin (T23366, Invitrogen) for 10 min at 37 °C. Holo-transferrin (500 μg/ml, T4132, Sigma) was added to controls. Uptake was stopped by washing twice with ice-cold acid wash (PBS supplemented with 150 mM NaCl and 20 mM citric acid, pH 5) and then with ice-cold RPMI + 0.5% BSA. Cells were stained for surface markers and analysed by flow cytometry.

**Kynurenine uptake**. HBSS was pre-warmed to 37 °C and kynurenine (final concentration 200 μM, K8525, Sigma), BCH (final 10 mM, A7902, Sigma), leucine (final 5 mM, L8912, Sigma) and lysine (final 5 mM, L8662, Sigma) was prepared. After surface antibody staining of samples, cells were resuspended in pre-warmed HBSS. Kynurenine and/or control reagents were added and samples were kept at 37 °C. The uptake was stopped after 5 min by adding 4% PFA for 30 min at room temperature, in the dark. Samples were measured at Canto II (BD, 405 nm laser, 450/50 BP filter for kynurenine fluorescence detection).

**Seahorse analysis of ECAR and OCR**. For real-time analysis of the extracellular acidification rate (ECAR) and oxygen consumption rate (OCR) of NK cells ex vivo, NK cells were isolated from spleens of naive and FV-infected mice. All wells were treated with Cell-Tak (734-1081, Corning) to ensure that the NK cells adhere to the plate. 200,000 isolated NK cells were plated per well and analysed by Seahorse XFe-96 Analyser (Agilent Technologies, Santa Clara, CA, USA). Sequential measurements of ECAR and OCR following the addition of the inhibitors (Sigma) oligomycin (2 μM, O4876), FCCP (1 μM, C2920), rotenone (100 nM, R8875) plus antimycin A (4 μM, A8674), and 2-deoxyglucose (2DG, 30 mM, 111980050, Acros Organics) enabled the calculation of basal glycolysis, glycolytic capacity, basal mitochondrial respiration, ATP production and maximal mitochondrial respiration.

**ATP determination**. NK cells were washed twice with PBS and lysed in Glo Lysis Buffer (E2661, Promega). ATP detection with ATP Determination Kit (A22066, Invitrogen) was done according to the manufacturer's instructions.

**NK cell isolation and depletion**. NK cell enrichment was done with MojoSort Mouse NK Cell Isolation Kit (480050, BioLegend) according to the manufacturer's instructions. NK cells were depleted by i.p. injections of 10 μg purified anti-NK1.1 depletion antibody (clone PK136, BE0036, BioXcell). The injection was done one day before FV infection and one day after infection.

**In vitro cytotoxicity assay**. Isolated, splenic NK cells ($25 \times 10^4$) from different groups were co-incubated with Tag-It Violet Tracking Dye (425101, BioLegend) stained YAC-1 and FBL-3 tumour cells ($1 \times 10^4$). The cytotoxic assay was performed in 96-well U-bottom plates. The cells were co-incubated for 18 h in a humidified 5% CO$_2$ atmosphere at 37 °C. Cells were washed once and stained with Zombie NIR (423106, BioLegend) to exclude dead cells. After a wash, cells were directly analysed by flow cytometry.

**Cytokine detection**. Cytokines were detected with a custom-made Legendplex kit (according to the manufacturer's instructions (BioLegend)).

**Transcriptome analysis**. RNA expression by array was reanalysed from GSE39555 using the Phantasus web platform version 1.12.0 (https://artyomovlab.wustl.edu/phantasus). Differential expression analysis was performed using LIMMA version 4.1. and significantly upregulated nutrient transporters and glycolysis-related genes were plotted.

**Real-time PCR**. Isolated NK cells were stored in DNA/RNA Shield (R1100, Zymoresearch). DNA was isolated with the Quick-DNA Microprep Kit (D3020, Zymoresearch). Real-time PCR was performed in duplicates using the PerfeCTa

SYBR Green Supermix (95054, Quanta Bio). Relative changes in DNA were calculated by ΔΔCt method: $2^{\wedge}-([Ct\ mtDNA_{FV} - Ct\ nDNA_{FV}]) - [(Ct\ mtDNA_{naive} - Ct\ nDNA_{naive})]$ (Supplementary Table 1).

RNA was extracted from liver explants harvested into RNA later (AM7021, ThermoFisher Scientific) and the RNeasy Plus kit (74034, Qiagen). RNA was transcribed to cDNA (4387406, Life Technologies) for quantitative PCR on the Applied Biosystems 6500 Fast Real-Time PCR system machine using TaqMan assays according to the manufacturer's protocols. Gene expression is presented as $2^{\wedge}([Ct\ of\ endogenous\ control\ gene] - [Ct\ of\ gene\ of\ interest])$. Hprt was used as an endogenous control gene for liver qRT-PCR (Mm01545399_m1, ThermoFisher).

**Iron depletion**. Mice were injected intraperitoneally with 100 nmol of mini-hepcidin PR73 (Ida-TH-Dpa-bhPro-RCR-bhPhe-Ahx-Ida(Hexadecylamine)-NH2) dissolved in 100 μl of DSPE (SUNBRIGHT DSPE-020CN, a PEG-phospholipid based solubilizer, NOF Corporation), whereas naive and FV-infected control groups were injected with the same amount of solvent. Injections were done every day at the same time.

**Analysis of serum iron**. Murine blood was placed in a BD microtainer SST tube (365968, Beckton Dickinson). Serum was obtained by spinning the clotted blood sample was spun at 8000×g for 5 min and stored at −80 °C. Serum iron was quantified using the Abbott Architect c16000 automated analyser (Abbott Laboratories) and the MULTGENT Iron Kit at Oxford John Radcliffe Hospital, UK, or at Atellica CH 930 Analyzer (Siemens Healthineers) and Iron_2 method at University Hospital Essen, Germany.

**In vitro NK cell culture**. Splenocytes were isolated from naive C57BL/6 mice and cultured for 6 days in 10 ng/ml IL-15. On day 4, iron availability was modified by adding DFO (4, 20, 100 μM, D9533, Sigma). On day 6, IL-2, IL-12 and FeSO₄ (200 μM, F8633, Sigma/Merck) was added, if indicated.

**Statistical analysis**. Statistics were calculated and graphs prepared with GraphPad Prism version 8 and Excel 2016. Statistical differences between two different groups were determined by Mann–Whitney test (nonparametric) or unpaired t-test (parametric). Analyses including several groups were tested using the nonparametric Kruskal–Wallis one-way analysis of variance (ANOVA) on ranks and Dunn's multiple-comparison test (nonparametric) or ordinary one-way ANOVA and Tukey multiple-comparison test (parametric). For the determination of correlating variables, linear regression and Goodness of fit were performed.

**Reporting summary**. Further information on research design is available in the Nature Research Reporting Summary linked to this article.

## Data availability

The authors declare that the data supporting the findings of this study are available within the paper. The source of RNA expression by the array (Affymetrix Mouse Genome 430 2.0 Array) is GSE39555. RNA expression by array data was accessed online: GSM9716290 [https://www.ncbi.nlm.nih.gov/geo/query/acc.cgi?acc=GSM971629], GSM971641, GSM971631, and GSM971643. Source data are provided with this paper.

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

## Acknowledgements
E.L.-S. is supported by a grant from the German Research Foundation (DFG) LI 3282/1-1. The DKFlab is supported by funding from the European research council (ERC-CoG 770769) and Science Foundation Ireland (18/ERCS/6005). C.C. is supported by an Irish Research Council Government of Ireland Postgraduate Scholarship (GOIPG/2018/2700). The Drakesmith lab is supported by the UK Medical Research Council.

## Author contributions
E.L.-S. and D.K.F. designed the study. E.L.-S., D.M., C.C., J.N.F., K.T.L., D.K.A., S.O.S. and B.W. performed the experiments. E.L.S and D.K.F. wrote the original draft of the manuscript. E.L.-S., D.K.F., U.D., D.M., H.D., J.N.F. and C.A.B. reviewed and edited the manuscript.

## Funding

## Competing interests
The authors declare no competing interests.
