## [Peer Review File · Nature Communications]

Metabolic requirements of NK cells during the acute response against retroviral infectionREVIEWER COMMENTS

Reviewer #1 (Remarks to the Author):

Review of Littwitz-Salomon et al: Metabolic requirements of NK cells during the response against retroviral infection

In this manuscript (MS) by Littwitz-Salomon et al, the authors show that NK cell's metabolic profile is altered upon FV infection in mice. The authors demonstrate that activated NK cells from infected mice increase metabolic activities including glycolysis, amino acid (AA) import and iron consumption. The authors show that when NK cells are unable to import AAs (in the absence of the Slc7a5 solute carrier) or are found in low iron environments, NK cell function is lacking. The field of activated NK cell metabolism is of growing interest in recent years and the importance of metabolic capacities to the function of immune cells upon infection has been demonstrated several times. The topic of the MS is of interest and the authors use elegant experiments to measure NK cell's metabolic capacities during acute infection.

It is unclear how the data presented in the MS regarding FV differ from what is known about NK cell function in other models of viral infections since there is no comparison in experiments to other models of infection. Additionally, although the authors demonstrate the importance of AA import as well as iron homeostasis for NK cell function, it is not shown to be physiologically relevant to host outcome from viral infection.

Comments to the authors:

- The authors present results supporting increased glycolysis and high OXPHOS in NK cells from infected mice. In the second result section the authors show increased levels of cMyc in NK cells of high FSC. cMyc was shown to drive cell proliferation through (among many other mechanisms) upregulation of pentose phosphate pathway (PPP, for the generation of nucleotides), supporting this pathway would mean taking away substrates of glycolysis. The authors should at least discuss this in the MS.
- The metabolic profile of NK cells was studied extensively in models of viral infections in both mice and humans and the role of iron was pointed out as crucial for NK cell function in these conditions. The role that the virus type (retroviruses) plays in this MS is unclear.
- The authors show that Kynurenine uptake in NK cells from infected mice is increased in both the spleen and the bone marrow (BM) and argue that this increase is due to higher levels of the transporter. In Figure 4c CD98 levels are shown in both spleen and BM of naive vs infected mice but although higher uptake is evident in both tissues, only splenic NK cells express higher levels of CD98 after infection. What is then the explanation behind the increased uptake of kynurenine in BM?
- The total number of Slc7a5 KO NK cells in the BM was shown not to be different from the WT in naive mice (Figure 5a). Upon infection, a difference in absolute numbers is evident between WT and KO mice (Figure 5e). The authors argue that this difference is due to impaired proliferation of NK cells but from the data shown in the figure it seems that NK cell numbers decrease in the KO compared to Naive mice rather than not catching up with the proliferation of WT NK cells. This might suggest cell death rather than impaired proliferation as an explanation for the difference in absolute numbers.
- In the KI-67 staining a reverse pattern is shown where both the KO and WT NK cells from infected mice have higher KI-67 staining as compared to NK cells from naive mice. If cells are positive for KI-67 then this should have also been evident in absolute numbers presented in Figure 5e.
- In lines 310-313, the authors write: these data show that the Slc7a5 transporter has an influence on NK cell numbers and cytotoxicity during viral infection but is dispensable for other aspects of the NK cell response including cytokine production and various metabolic parameters. In the beginning of the MS the authors use a panel of metabolic assays to determine the differences in the metabolic profile of NK cells from naive vs infected mice whereas here the statement about these various metabolic parameters not being any different as a result of Slc7a5 KO is solely based on cMyc expression levels. This statement is insufficiently supported by the data. Thus, the authors should further demonstrate that this is indeed the case for Slc7a5 KO NK cells or remove this conclusion from the text.
- It is unclear why the authors choose to include SARS-CoV-2 along the MS introduction and result

sections where no data is shown regarding SARS-CoV-2 and the title clearly states that the focus is on retroviruses.

Minor comments:

- It is generally recommended not to use “extreme” wordings such as “nothing”, “never”, etc. when reviewing the body of literature regarding the field or phenomenon discussed in one’s manuscript. One can never be certain that they covered every possible corner of existing publications and make such a confident statement. Moreover, a quick and superficial literature search brings up several papers (and some preprints) discussing NK cell metabolic profiling during retroviral infection (see for example Costanzo MC et al, published in March 2018 in this very journal). Although the work described in the MS is novel and important, it certainly is not the first of its kind in the field. Thus, any statement of an “extreme” character should be modified according to the logic presented above.
- Additionally, in lines 400-401 the authors claim that this is the first time it has been demonstrated that NK cells have increased metabolic activity in response to viral infection in vivo, a statement which is not supported by the current body of literature.
- There are several mistakes and grammatical inaccuracies along the MS, the authors should go over the MS again and fix them (examples: Line 106: “...that is been investigated...”, lines 154-155 “... 7 after infection...”, line 209 “... an metabolically active...”)
- Result section titles should be(consistently) written in present tense.
- It is unclear why the authors define an acronym such as 7dpi (for days post infection) and then not use it consistently along the MS. Acronyms, when used moderately can make the reading experience easier and more fluent.
- Figure 4b (right panel) should be fixed.

Reviewer #2 (Remarks to the Author):

Manuscript Nr: NCOMMS-20-51020

Littwitz-Salomon et al., “Metabolic requirements of NK cells during the response against retroviral infection”

The authors demonstrate that NK cells expand during Friend retrovirus (FV) infection and acquire metabolic changes that are consistent with NK cell activation. Compromising amino acid uptake by Slc7a5 does not seem to affect most NK cell effector functions except for cytotoxicity which is measured against YAC-1 and FV derived tumor cells. In contrast to this rather modest phenotype, hypoferremia induced by mini-hepcidin affects both NK cell derived cytokine production and their cytotoxicity. This stronger inhibition of NK cell function is then also associated with elevated numbers of FV infected cells in the bone marrow. However, this effect might not be exclusively due to functional differences in the NK cell compartment.

Therefore, the functional consequences of the metabolic NK cell manipulations for FV infection are somewhat underwhelming and should be investigated in more detail.

Major comments:

1. The authors report changes in NK cell differentiation (CD11b and CD27) during FV infection. Do also tissue resident NK cells (CXCR6, CD49a) develop in the bone marrow and spleen of FV infected mice?
2. The authors seem to observe differences in cytokine production by NK cells during FV infection and hypoferremia, but to a lesser extent when Slc7a5 was missing. Is this however relevant for the overall cytokine production during infection. Are cytokine levels in serum altered in the absence of Slc7a5 on NK cells and during hypoferremia?
3. The authors demonstrate that Slc7a5 deficiency reduces cytotoxicity of NK cells during FV infection but does this influence the outcome of the infection. Are FV viral titers or number of infected cells changed due to Slc7a5 deficiency in NK cells?
4. Why was cytotoxicity during iron depletion not measured against YAC-1 cells?
5. The authors record cytotoxicity and cytokine production during FV infection in iron deficient mice. Which of these NK cell functions is protective during FV infection?
6. Finally the only indication that the observed changes in the NK cell compartment upon

metabolism manipulation affect FV infection are provided in the last figure. However, are NK cells responsible for the decreased immune control of FV infection in the bone marrow during hypoferremia (figure 7D)? Does iron depletion with mini-hepcidin no longer affect the number of FV infected cells in the bone marrow upon NK cell depletion? An additional read-out like viral loads or histochemistry could also strengthen this analysis.

7. It is unclear how the first part of the manuscript (role of amino acid import for NK cell function during viral infections) is connected to the second part (iron requirement for NK cell function). The authors try to link the two parts via CD71, its regulation by cMyc and the role of amino acid import for cMyc regulation. However, the functional consequences of Slc7a5 deficiency and hypoferremia seem significantly different, especially with respect to cytokine production. This should at least be discussed.

Minor comments:

1. Line 163, activation instead of actiation

In summary, the authors report a variety of changes in NK cells upon FV infection. These are interesting, but their functional relevance for FV infection remains unclear. For the specific Slc7a5 deficiency in NK cells no viral parameters are reported, and for hypoferremia the dependence of altered infection on NK cells was not analyzed. If the observed changes do not alter immune control of FV their significance and advance over previously published information remains modest.

Reviewer #3 (Remarks to the Author):

The manuscript by Littwitz-Salomon et al. reveals that NK cells have metabolic requirements during acute retroviral infection and they are important for antiviral immunity. They used a very nice Friend retrovirus (FV) mouse model to analyze metabolic changes in NK cells against retrovirus infection. Remarkably, although mechanisms implicated in antiviral NK cell response have been studied, nothing was known about NK cell metabolism in the acute phase of infection. Here, the authors provide a nice set of experiments showing that metabolic changes in NK cells occur during FV infection through the increase of glycolysis and mitochondrial metabolic pathways. NK cells also increase nutrient uptake such as amino acids (kynurenine) and iron. Specific deletion of the amino acid transporter Slc7a5 reduced NK cytotoxicity and iron deficiency affects NK cell antiviral functions showing the crucial role of metabolites in NK functions upon FV infection. Overall, this is an impactful study that provides important and novel insights into how metabolic factors are implicated in antiviral NK T cell response and highlights a novel metabolism-target approaches for treating infectious diseases.

Still, major problems need to be clarified and crucial missing experiments to complete the data need to be realized before publication.

*Major points for further consideration:

-In Fig 1c, authors show that FV infection induces a decrease of immature (CD27-CD11b-) and cytotoxic effector cells (CD27-CD11b+) and an increase of immature (CD27+CD11b+) and mature cytokine producers (CD27+CD11b-). However, the analysis of Fig. 1d-e has been done on total NK cells. I supposed that the difference in IFN- γ , TNF- α , FasL, Gzmb levels in total NK cells is related to mature cytokine NK producers but it has to be shown.

-Fig 3: Excepted for the Seahorse, which requires a lot of cells, the authors could go further by analyzing which NK populations undergo these metabolic changes during infection.

-Littwitz-Salomon et al. show an augmented glycolytic activity in NK cells from FV-infected mice which can be due to an increase glucose uptake. To complement the data showed in Fig. 3g-h, it would be great to assess glucose uptake of NK cells in naïve vs FV-infected animals (for example with 2-NBDG).

-Fig. 4 is very light and many experiments must to be done to complete the data:

*LAT1 preferentially transport branched-chain (valine, leucine, isoleucine) and aromatic (tryptophan, tyrosine) amino acids. NK cells could also uptake some of these specific amino acids in addition to kynurenine? (Some specific assays are available or metabolomic analysis).
*Again, it would be very nice to know which NK subpopulations uptake kynurenine.
*Authors demonstrated that NK cells increase kynurenine following FV infection but an experiment showing the direct effect of kynurenine on NK (function?) during infection is critical.

-In Fig 5, a lot of experiments has to be done:

*It would be appreciated to include for each experiment Slc7a5NK-KO at steady state.
*In Fig 5e, authors showed that specific depletion of Slc7a5 leads to a decrease in NK cell numbers in BM and spleen. Once again, a histogram showing all NK subpopulations in order to know which NK subsets is affected by this depletion is missing.
*Authors concluded that Slc7a5 transporter is dispensable for NK metabolism. The conclusion is dangerous given that authors have analyzed only few markers related to metabolism (CD71 and Myc). A Seahorse analysis of NK cells from Slc7a5NK-WT vs Slc7a5NK-KO at steady state and upon FV infection will reinforce this light conclusion.

-In Fig. 7, Littwitz-Salomon et al. show that iron deprivation inhibits NK cell responses against FV infection. However, an analysis of viral load during 7 days (as shown in Fig. 1a) between mice injected or not with mHep is missing and important.
Furthermore, iron is also uptake by other immune cells. How can we be sure that the mHep injection has a direct effect on NK? Authors might show the direct effect of iron uptake on NK cells and their antiviral responses (in vitro experiments?)

-Finally, an experiment where the addition of kynurenine or iron (or both) ameliorates NK function and boosts their antiviral response is missing and required.

*Minor issues:

-In the introduction, references must be added (lines 48 and 62)

-Line 163: "v" is missing for "NK cell activation"

-In the 4th part of the text, the title "NK cells increase amino acid uptake following FV infection" should be replace by "NK cells increase KYNURENINE uptake following FV infection" since no other amino acids have been shown in this part.

-In Fig 2, the authors wanted to know if acute FV infection induces metabolic changes in NK cells. Fig 2a-c have to be moved as Supplementary figures. It's out of context. The increase size is not always correlated with metabolic changes.

-Fig 2: To really know if acute FV infection induces metabolic changes in NK cells, it would help to actually do a comparison of markers (CD98, CD71) between naïve vs FV mice and between the different subpopulations of NK cells.

-Fig 5b axis labeling is missing

-Fig 5h: In the paper from Loftus et al. Nat Com, 2018, the authors showed that Slca7a5 controls cMyc levels in NK cells. As control, it would be great to add the expression of cMyc in Slc7a5NK-WT vs Slc7a5NK-KO at steady state (as control).

-Fig 6 and Fig 7 showing that iron availability is important for NK cells during infection and that iron deprivation inhibits NK response should be combined in a single figure.

-Most graphs show NK frequency, whereas in fig 5e, 5f, histograms represent NK number. Frequencies in these figures or frequencies and numbers for every panels are required.

Please find below our point-by-point responses to the reviewers concerns in red text.

Reviewer #1 (Remarks to the Author):

Review of Littwitz-Salomon et al: Metabolic requirements of NK cells during the response against retroviral infection

In this manuscript (MS) by Littwitz-Salomon et al, the authors show that NK cell's metabolic profile is altered upon FV infection in mice. The authors demonstrate that activated NK cells from infected mice increase metabolic activities including glycolysis, amino acid (AA) import and iron consumption. The authors show that when NK cells are unable to import AAs (in the absence of the Slc7a5 solute carrier) or are found in low iron environments, NK cell function is lacking.

The field of activated NK cell metabolism is of growing interest in recent years and the importance of metabolic capacities to the function of immune cells upon infection has been demonstrated several times.

While there are some papers that have studied some minor aspects of NK cell metabolism in the context of viral infection there is very little information about the metabolic changes that accompany NK cells responding to acute viral infection. Here is a summary of the relevant data:

Glycolytic requirement for NK cell cytotoxicity and cytomegalovirus control.

Mah AY, Rashidi A, Keppel MP, Saucier N, Moore EK, Alinger JB, Tripathy SK, Agarwal SK, Jeng EK, Wong HC, Miller JS, Fehniger TA, Mace EM, French AR, Cooper MA.

Show that 2DG treated mice have an increase susceptibility to MCMV infection

No measurements of metabolic fluxes of NK cells during viral infection

No measurements of metabolic signalling at all.

BNIP3- and BNIP3L-Mediated Mitophagy Promotes the Generation of Natural Killer Cell Memory.

O'Sullivan TE, Johnson LR, Kang HH, **Sun JC**. Immunity. 2015 Aug 18;43(2):331-42. doi: 10.1016/j.immuni.2015.07.012. Epub 2015 Aug

Track some mitochondrial parameters during the course of MCMV infection.

No measurements of metabolic fluxes

No measurements of metabolic signalling.

IL-15 re-programming compensates for NK cell mitochondrial dysfunction in HIV-1 infection

Elia Moreno Cubero, Stefan Balint, Aljawharah Alrubayyi, Ane Ogbe, Rebecca Matthews, Fiona Burns, Sarah Rowland-Jones, Persephone Borrow, Anna Schurich, Michael Dustin, Dimitra Peppas

doi: <https://doi.org/10.1101/811117>

BioRxiv

Show decreases in metabolic fluxes in dysfunctional NK cells from patients with HIV1 (chronic disease rather than acute infection). Also, some mitochondrial changes.

Does not study NK cell metabolic response during retroviral infection but rather NK cell dysfunction in chronic disease.

Metabolic but not transcriptional regulation by PKM2 is important for natural killer cell responses.

Walls JF, Subleski JJ, Palmieri EM, Gonzalez-Cotto M, Gardiner CM, McVicar DW, **Finlay DK**. *Elife*. 2020 Aug 19;9:e59166. doi: 10.7554/eLife.5916

Used the MCMV model but did no metabolic measurements and did not look at metabolic signalling.

ARID5B regulates metabolic programming in human adaptive NK cells.

Cichocki F, Wu CY, Zhang B, Felices M, Tesi B, Tuininga K, Dougherty P, Taras E, Hinderlie P, Blazar BR, Bryceson YT, Miller JS. *J Exp Med*. 2018 Sep 3;215(9):2379-2395. doi: 10.1084/jem.20172168. Epub 2018 Jul 30

Study metabolic changes in adaptive (or memory) NK cells in patients that are seropositive for CMV – i.e had been infected with CMV at some point. Did not study how the metabolic requirements of NK cells responding to virus.

The critical role of IL-15–PI3K–mTOR pathway in natural killer cell effector functions

Neethi Nandagopal, Alaa Kassim Ali, Amandeep Kaur Komal and Seung-Hwan Lee. *Front Immunol*. 2014 Apr 23;5:187. doi: 10.3389/fimmu.2014.00187. eCollection 2014.

Study shows that mTOR is important for the antiviral function of NK cells in MCMV infection. Systemic rapamycin treatment of mice. Only two figures of the manuscript contain MCMV data.

The metabolic checkpoint kinase mTOR is essential for IL-15 signaling during the development and activation of NK cells

Antoine Marçais Julien Cherfils-Vicini, Charlotte Viant , Sophie Degouve, Sébastien Viel, Aurore Fenis, Jessica Rabilloud, Katia Mayol, Armelle Tavares, Jacques Bienvenu , Yann-Gaël Gangloff , Eric Gilson, Eric Vivier , Thierry Walzer *Nat Immunol*. 2014 Aug;15(8):749-757. doi: 10.1038/ni.2936. Epub 2014 Jun 29.

mTOR is important for proliferation, GzmB and IFN γ after MCMV infection. Only a part of one figure contain MCMV data.

We have discussed the papers above that are directly relevant to the narrative of our study on acute viral infection in the discussion section. We are willing to expand this discussion to include chronic viral infection and more diverse signalling pathways, if the reviewers deem this appropriate, but we are keen to minimise any unnecessary complexity to ensure that this story is accessible to a wide readership.

The topic of the MS is of interest and the authors use elegant experiments to measure NK cell's metabolic capacities during acute infection.

It is unclear how the data presented in the MS regarding FV differ from what is known about NK cell function in other models of viral infections since there is no comparison in experiments to other models of infection.

As can be seen from the summary of the literature relating to NK cell metabolism a viral infection there is very little known about NK cell metabolism during the response to an acute viral infection in any models. To strengthen our conclusions, we now include some data on the MCMV model that shows increased NK cell metabolism (cell size, CD71, CD98, cMyc mRNA, Slc7a5 mRNA etc) during the acute phase of this viral infection (see new Figure 4)

Additionally, although the authors demonstrate the importance of AA import as well as iron homeostasis for NK cell function, it is not shown to be physiologically relevant to host outcome from viral infection.

We thank the reviewer for this comment. We have included new data showing the physiological role of Slc7a5 and serum iron deficiency after FV infection into the manuscript (see figure 5 and 7).

Comments to the authors:

- The authors present results supporting increased glycolysis and high OXPHOS in NK cells from infected mice. In the second result section the authors show increased levels of cMyc in NK cells of high FSC. cMyc was shown to drive cell proliferation through (among many other mechanisms) upregulation of pentose phosphate pathway (PPP, for the generation of nucleotides), supporting this pathway would mean taking away substrates of glycolysis. The authors should at least discuss this in the MS.

We have included some discussion of how cMyc can support NK cell biosynthetic processes and so proliferative capacity through glycolysis and PPP (lines 190-195).

- The metabolic profile of NK cells was studied extensively in models of viral infections in both mice and humans and the role of iron was pointed out as crucial for NK cell function in these conditions. The role that the virus type (retroviruses) plays in this MS is unclear.

To address the role of the virus type, we have now included some data on the metabolic changes within NK cells that accompany the acute phase of MCMV infection (see Figure 4 and below).

Data from Figure 4 showing flow cytometry analysis of NK cells on day 1.5 post MCMV infection

Therefore, NK cell metabolic responses are not restricted to retroviral infections and are likely to be broadly applicable to acute NK cell responses in viral infections. These are novel insights as the current literature has not studied metabolic response except mTOR during the acute phase of viral responses (see literature summary above). Most other studies have looked at NK cell dysfunction or NK cell memory following chronic viral infection.

- The authors show that Kynurenine uptake in NK cells from infected mice is increased in both the spleen and the bone marrow (BM) and argue that this increase is due to higher levels of the transporter. In Figure 4c CD98 levels are shown in both spleen and BM of naïve vs infected mice but although higher uptake is evident in both tissues, only splenic NK cells express higher levels of CD98 after infection. What is then the explanation behind the increased uptake of kynurenine in BM?

While CD98 (also called Slc3a2) forms a complex with a number of amino acid transporters it is not the pore forming subunit. The subunit that actually transports amino acids (and kynurenine is Slc7a5) and the increased kynurenine uptake is reflective of increased expression of Slc7a5. There are no good antibodies that detect Slc7a5 protein making the analysis of this protein by flow cytometry impossible. This is why the Kyn uptake assay is so useful. Analysis of available RNAseq datasets show that Slc7a5 mRNA is increased on NK cells during acute MCMV expression (see Figure 4 and below)

Data from Figure 4 showing analysis of RNAseq data of NK cells on day 1.5 post MCMV infection

- The total number of Slc7a5 KO NK cells in the BM was shown not to be different from the WT in naïve mice (Figure 5a). Upon infection, a difference in absolute numbers is evident between WT and KO mice (Figure 5e). The authors argue that this difference is due to impaired proliferation of NK cells but from the data shown in the figure it seems that NK cell numbers decrease in the KO compared to Naïve mice rather than not catching up with the proliferation of WT NK cells. This might suggest cell death rather than impaired proliferation as an explanation for the difference in absolute numbers.

This is a good point and the total cell number within an organ can be related to a number of factors including proliferation, death, and migration in the organ in question. We have now included data showing that there is no change in NK cell viability in the WT vs KO mice (Supplementary Figure 3 and below) that argues that cell death is not the underlying reason. In the discussion section we have discussed the other potential reasons that might explain these differences in total numbers.

Data from Supplementary figure 3b showing no differences in the viability of NK cell from FV-infected Slc7a5^{NK-WT} and Slc7a5^{NK-KO}.

- In the KI-67 staining a reverse pattern is shown where both the KO and WT NK cells from infected mice have higher KI-67 staining as compared to NK cells from naïve mice. If cells are positive for KI-67 then this should have also been evident in absolute numbers presented in Figure 5e.

This likely reflects the fact that absolute numbers of NK will also be affected by trafficking in and out of the BM and or spleen and cannot simply determined by the frequency of proliferation and/or cell death.

- In lines 310-313, the authors write: these data show that the Slc7a5 transporter has an influence on NK cell numbers and cytotoxicity during viral infection but is dispensable for other aspects of the NK cell response including cytokine production and various metabolic parameters. In the beginning of the MS the authors use a panel of metabolic assays to determine the differences in the metabolic profile of NK cells from naïve vs infected mice whereas here the statement about these various metabolic parameters not being any different as a result of Slc7a5 KO is solely based on cMyc expression levels. This statement is insufficiently supported by the data. Thus, the authors should further demonstrate that this is indeed the case for Slc7a5 KO NK cells or remove this conclusion from the text.

This is a very reasonable comment. In an effort to provide a clear narrative, we may have included too little data here. We now include additional metabolic data into supplementary figure 2 showing that there are no differences in other metabolic parameters including mitochondrial mass and polarization, transferrin uptake as measure for iron uptake and CD98 expression. From our experience, working on immunometabolism of a range of immune cell types the changes in cell size of an activated NK cells (or indeed T cells) usually correlates very well with the metabolic responses.

- It is unclear why the authors choose to include SARS-CoV-2 along the MS introduction and result sections where no data is shown regarding SARS-CoV-2 and the title clearly states that the focus is on retroviruses.

We are referencing a recent publication that showed that SARS-CoV-2 infection result in hypoferrremia similar to the altered iron levels HIV infection. It just serves as an example for reduced iron levels caused by virus infection.

Minor comments:

- It is generally recommended not to use “extreme” wordings such as “nothing”, “never”, etc. when reviewing the body of literature regarding the field or phenomenon discussed in one’s manuscript. One can never be certain that they covered every possible corner of existing publications and make such a confident statement. Moreover, a quick and superficial literature search brings up several papers (and some preprints) discussing NK cell metabolic profiling during retroviral infection (see for example Costanzo MC et al, published in March 2018 in this very journal). Although the work described in the MS is novel and important, it certainly is not the first of its kind in the field. Thus, any statement of an “extreme” character should be modified according to the logic presented above.

I agree with this point in principle and we have toned down the language so that it is not absolute.

That said I would say that the paper mentioned by Costanzo MC et al has some transcriptomic data that includes some changes in metabolic pathways. They do not study metabolic responses at all beyond this.

- Additionally, in lines 400-401 the authors claim that this is the first time it has been demonstrated that NK cells have increased metabolic activity in response to viral infection in vivo, a statement which is not supported by the current body of literature.

We apologise that this sentence was not precise enough. We have modified it from: “In this study, we demonstrate for the first time that there is an increased metabolic activity of NK cells responding to viral infection in vivo, in this case to acute Friend retrovirus (FV) infection” ...to“In this study, we demonstrate for the first time that there is an increased metabolic reprogramming and nutrient uptake of NK cells responding to acute viral infections in vivo, in this case to acute Friend retrovirus (FV) and murine cytomegalovirus (MCMV) infection”. The review of the literature above shows that no other group has looked at metabolic activity except for mTOR in NK cells during acute viral response and only a few have looked at NK cell metabolism during chronic viral infection. We hope that the inclusion of the work acute makes this sentence acceptable.

- There are several mistakes and grammatical inaccuracies along the MS, the authors should

go over the MS again and fix them (examples: Line 106: "...that is been investigated...", lines 154-155 "... 7 after infection...", line 209 "... an metabolically active...")

Thank you for spotting these errors. We apologise and have corrected them now.

- Result section titles should be (consistently) written in present tense.

Thank you for spotting this. We have now corrected this.

- It is unclear why the authors define an acronym such as 7dpi (for days post infection) and then not use it consistently along the MS. Acronyms, when used moderately can make the reading experience easier and more fluent.

Thank you for spotting this. We have now corrected this.

- Figure 4b (right panel) should be fixed.

Thank you for spotting this error. We have fixed it in the revised manuscript

Reviewer #2 (Remarks to the Author):

Manuscript Nr: NCOMMS-20-51020

Littwitz-Salomon et al., "Metabolic requirements of NK cells during the response against retroviral infection"

The authors demonstrate that NK cells expand during Friend retrovirus (FV) infection and acquire metabolic changes that are consistent with NK cell activation. Compromising amino acid uptake by Slc7a5 does not seem to affect most NK cell effector functions except for cytotoxicity which is measured against YAC-1 and FV derived tumor cells. In contrast to this rather modest phenotype, hypoferremia induced by mini-hepcidin affects both NK cell derived cytokine production and their cytotoxicity. This stronger inhibition of NK cell function is then also associated with elevated numbers of FV infected cells in the bone marrow. However, this effect might not be exclusively due to functional differences in the NK cell compartment.

Therefore, the functional consequences of the metabolic NK cell manipulations for FV infection are somewhat underwhelming and should be investigated in more detail.

We have addressed the question of whether the altered viral loads in mice treated with mHep is due to altered function within the NK compartment in a number of ways.

(1) We considered whether there might be a defect in the production of the cytokines (IL-15 and IL-18), by macrophages and dendritic cells, that are known to activate NK cells in response to FV virus infection. While there were some minor changes in the activation of macrophages and dendritic cells in mice infected with FV and administered mHep, the key point is that there was no decrease in the abundance of IL-15 or IL-18 levels measured in the serum and the spleens. In fact, there were increased levels of IL-12 and IL-18 and a trend towards increased levels of IL-15 in the spleen of mHep treated mice. (Figure 6j and below)

Data from figure 6j. Cytokine concentrations measured in the spleen of FV-infected and FV-infected, mHep-treated groups.

(2) Our previous work showed that if NK cells are depleted from the mouse (injection of NK1.1 antibody) there is an increase in FV-infected cells - published in JV in 2017 doi: 10.1128/JVI.01122-17.

Figure from JV paper (doi: 10.1128/JVI.01122-17) showing the increased viral loads after depletion of NK cells in acute FV infection.

We reasoned it NK cells were dysfunctional in mHep-treated mice following FV infection then removing NK cells would have no additional effect on viral loads. Indeed, when we depleted NK cells from the mice through the injection of anti-NK1.1 prior to infection with FV + mHep treatment, mice lacking NK cells showed no increase in virus-infected cells. (Figure 7i and j, and below)

Data from figure 7i and 7j showing the depletion efficiency of NK cells and the viral loads after mHep-treatment and NK cell depletion.

(3) We considered the direct impact of reduced iron availability on NK cell response. In these experiments NK cells were activated ex vivo in the presence of increasing concentrations of an iron chelator and the impact on NK cell growth (cell size, FSC-A), NK cell metabolism (mitochondrial mass) and NK cells function (perforin and granzyme B expression) were investigated. All these parameters were reduced in the presence of the iron chelator in a dose dependent manner and importantly they were restored with the additional supplementation of the cultures with FeSO₄. (Figure 7e-h and below)

Data from figure 7 e-h. In vitro cultures of NK cells and treatment with the iron chelator DFO resulted in a dose dependent reduction of cell size, MitoTrackerGreen and Perforin as well as Granzyme B expression. The NK cell phenotype could be rescued by the addition of iron ($FeSO_4$).

We feel that taken together these new data make a strong argument that limited iron availability affects FV infection largely through the inhibition of NK cell metabolism and function. We thank the reviewer for these suggestions.

Major comments:

1. The authors report changes in NK cell differentiation (CD11b and CD27) during FV infection. Do also tissue resident NK cells (CXCR6, CD49a) develop in the bone marrow and spleen of FV infected mice?

Tissue-resident NK cells represent a very small proportion of NK cells in these organs but they do also increase in number in FV infected mice (see below). We feel that including this data would distract from the focus of the manuscript and so do not propose including it in the final manuscript. However, we are happy to be guided by the editors and reviewer 2 on this matter.

Figure R1: Tissue-resident NK cells after acute FV infection.

C57BL/6 mice were infected with 40000 SFFU of FV or used as naïve controls. NK cells were analysed at 7 dpi in spleen and bone marrow. Tissue residency was analysed by the expression of CD49a and CXCR6 on the NK cell population. Statistically significant differences were analysed within the organs with an unpaired *t* test. A minimum of six animals were used for the analysis.

2. The authors seem to observe differences in cytokine production by NK cells during FV infection and hypoferremia, but to a lesser extent when Slc7a5 was missing. Is this however relevant for the overall cytokine production during infection. Are cytokine levels in serum altered in the absence of Slc7a5 on NK cells and during hypoferremia?

We have performed an analysis of IFN γ levels in the serum of mice during FV virus infection. We see an increase in IFN γ production in the serum of mice infected for 7 days with FV that is completely abolished by mHep treatment. These results are in line with the IFN γ mRNA molecule analysis in the spleen (original Figure 5i). After FV-infection, we see an increase in IFN γ levels in the serum of WT FV-infected mice compared to naïve IFN γ

concentrations, but there is no significant difference in KO mice (Supplementary Figure 3 and below) as it is shown for NK cells in the histogram (Figure 5g).

Data from supplementary figure 3d shows no difference in the IFN γ concentration between the Slc7a5^{NK-WT} and Slc7a5^{NK-KO} mice.

In terms of other cytokines, there is no decrease in IL-12, IL-15 or IL-18 in the spleens of mice infected with FV and treated with mHep (shown above).

3. The authors demonstrate that Slc7a5 deficiency reduces cytotoxicity of NK cells during FV infection but does this influence the outcome of the infection. Are FV viral titers or number of infected cells changed due to Slc7a5 deficiency in NK cells?

We have looked at viral infected cells on day 3 and day 7 but there are no significant differences in KO mice (Figure 5j and below) This data again suggests that NK cells have some ability to adapt in vivo to the lack of Slc7a5 expression (we talk about this in the discussion section).

Data from figure 5j shows no differences in viral loads in bone marrow and spleen between the FV-infected Slc7a5^{NK-WT} (grey) and Slc7a5^{NK-KO} (red) mice.

4. Why was cytotoxicity during iron depletion not measured against YAC-1 cells?

We did measure cytotoxicity against YAC-1 cells in hypoferremia but due to interexperiment variation there was no significant decrease though it was clear in each experiment that cytotoxicity was reduced in each individual experiment (see below). Considering that the FBL-3 cells are a much more physiological relevant target cells (FV-induced cells that show FV antigens on surface) during FV infection, we were confident that mHep administration reduces NK cell cytotoxicity.

Figure R2: Yac1 tumor cell killing by NK cells. Mice were infected with FV and treated with vehicle or vehicle+mHep. Yac1 cells were stained with Tag-It violet tracking dye and co-cultured with isolated NK cells. Co-culture was stained for viability and measured at flow cytometer. Statistically significant differences were analysed by Mann-Whitney test.

5. The authors record cytotoxicity and cytokine production during FV infection in iron deficient mice. Which of these NK cell functions is protective during FV infection?

During FV infection, NK cells are important for the control of early viral replication (Littwitz et al., 2013, doi: 10.1186/1742-4690-10-127; Littwitz-Salomon et al., 2017, doi:

10.1128/JVI.01122-17). Depletion of NK cells results in increased viral loads as we already published. In the very acute FV infection, molecules related to cytotoxicity such as death receptor ligands but also granzymes are expressed and produced by activated NK cells. At 7 dpi, we observed significantly increased levels of cytokines such as IFN γ compared to naïve mice. NK cells from mHep-treated, FV-infected mice express similar levels of IFN γ to naïve mice illustrating the dysfunction of NK cells after serum iron reduction with mHep.

Interestingly, IFN γ has direct antiviral functions as it was shown by multiple other groups. We also did an additional analysis of FasL and Perforin and observed significant reductions between FV (red line) and FV-infected and mHep-treated (blue lines) groups (below). Thus, we believe that eliminating virus-infected target cells require multiple antiviral mechanisms such as cytokines and cytotoxic molecules.

Figure R3: Representative histograms of Perforin and FasL expression of NK cells after FV infection. Mice were infected with FV and one group treated with mHep injections i.p. every day. Cells were analysed for FasL and Perforin as it is displayed.

6. Finally the only indication that the observed changes in the NK cell compartment upon metabolism manipulation affect FV infection are provided in the last figure. However, are NK cells responsible for the decreased immune control of FV infection in the bone marrow during hypoferremia (figure 7D)? Does iron depletion with mini-hepcidin no longer affect the number of FV infected cells in the bone marrow upon NK cell depletion? An additional read-out like viral loads or histochemistry could also strengthen this analysis.

We have discussed this in detail above. Briefly, NK cell depletion in mHep-treated FV-infected mice did not result in increased viral loads, as it was published from us before in FV-infected and NK cell-depleted mice (Littwitz-Salomon et al., 2017). So we conclude that NK cell antiviral functions are impaired without serum iron (mHep group).

7. It is unclear how the first part of the manuscript (role of amino acid import for NK cell function during viral infections) is connected to the second part (iron requirement for NK cell function). The authors try to link the two parts via CD71, its regulation by cMyc and the role of amino acid import for cMyc regulation. However, the functional consequences of Slc7a5 deficiency and hypoferremia seem significantly different, especially with respect to cytokine production. This should at least be discussed.

We now include a discussion of this in the manuscript

Minor comments:

1. Line 163, activation instead of actiation

Thank you for spotting this. We have now corrected this.

In summary, the authors report a variety of changes in NK cells upon FV infection. These are interesting, but their functional relevance for FV infection remains unclear. For the specific Slc7a5 deficiency in NK cells no viral parameters are reported, and for hypoferremia the dependence of altered infection on NK cells was not analyzed. If the observed changes do not alter immune control of FV their significance and advance over previously published information remains modest.

Thank you for your fair and constructive comments. We hope that we have addressed them satisfactorily in our revised manuscript.

Reviewer #3 (Remarks to the Author):

The manuscript by Littwitz-Salomon et al. reveals that NK cells have metabolic requirements during acute retroviral infection and they are important for antiviral immunity. They used a very nice Friend retrovirus (FV) mouse model to analyze metabolic changes in NK cells against retrovirus infection. Remarkably, although mechanisms implicated in antiviral NK cell response have been studied, nothing was known about NK cell metabolism in the acute phase of infection.

Here, the authors provide a nice set of experiments showing that metabolic changes in NK cells occur during FV infection through the increase of glycolysis and mitochondrial metabolic pathways. NK cells also increase nutrient uptake such as amino acids (kynurenine)

and iron. Specific deletion of the amino acid transporter Slc7a5 reduced NK cytotoxicity and iron deficiency affects NK cell antiviral functions showing the crucial role of metabolites in NK functions upon FV infection.

Overall, this is an impactful study that provides important and novel insights into how metabolic factors are implicated in antiviral NK T cell response and highlights a novel metabolism-target approaches for treating infectious diseases.

Still, major problems need to be clarified and crucial missing experiments to complete the data need to be realized before publication.

*Major points for further consideration:

-In Fig 1c, authors show that FV infection induces a decrease of immature (CD27-CD11b-) and cytotoxic effector cells (CD27-CD11b+) and an increase of immature (CD27+CD11b+) and mature cytokine producers (CD27+CD11b-). However, the analysis of Fig. 1d-e has been done on total NK cells. I supposed that the difference in IFN- γ , TNF- α , FasL, Gzmb levels in total NK cells is related to mature cytokine NK producers but it has to be shown.

The analysis of CD69, IFN γ , TNF α and GzmB was done in mature NK cells (CD49b⁺NK1.1⁺ cells). Nevertheless, we now include the production/expression of these parameters in the different NK cell subsets in supplementary figure 1 and below.

Data from supplementary figure 1 showing the subset distribution (CD27 CD11b) within the NK cell population for activation (CD69) and effector functions such as IFN γ , TNF α and Granzyme B.

-Fig 3: Excepted for the Seahorse, which requires a lot of cells, the authors could go further by analyzing which NK populations undergo these metabolic changes during infection.

We do have information on the different subsets that we can include in the manuscript but we are reticent to add it to the main body of the manuscript. We feel that it would complicate and distract the narrative for non-NK cell expert readers. In summary we do see differences in cytokine production and metabolic parameters in the different subsets though how this relates to the overall anti-viral response is not clear and we feel beyond the scope of this current manuscript. These data are now in supplementary figure 1 and below.

Data from supplementary figure 1 showing the subset distribution (CD27 CD11b) within the NK cell population for nutrient receptors (CD98, CD71), the transcription factor cMyc and transferrin.

-Littwitz-Salomon et al. show an augmented glycolytic activity in NK cells from FV-infected mice which can be due to an increase glucose uptake. To complement the data showed in Fig. 3g-h, it would be great to assess glucose uptake of NK cells in naïve vs FV-infected animals (for example with 2-NBDG).

Unfortunately, the only way to measure glucose uptake into cells is using radiolabelled glucose and requires millions of purified cells and so was not possible in this study. While 2-NBDG has been used by a lot of groups in the past to measure glucose uptake by flow cytometry, it has recently been proven that this assay does not measure glucose uptake at all but rather is a somewhat non-specific measure of cell size (Sinclair et al.

Immunometabolism.2020;2(4):e200029. <https://doi.org/10.20900/immunometab20200029>)

-Fig. 4 is very light and many experiments must to be done to complete the data:

*LAT1 preferentially transport branched-chain (valine, leucine, isoleucine) and aromatic (tryptophan, tyrosine) amino acids. NK cells could also uptake some of these specific amino acids in addition to kynurenine? (Some specific assays are available or metabolomic analysis).

Yes, absolutely, Slc7a5 does transport lots of different amino acids and this is why it is important. We are not making the point that kynurenine is in anyway important in the response to FV, kynurenine is just a convenient naturally fluorescent cargo that allow us to get a measure of Slc7a5 expression and activity (there is no Slc7a5 antibody available that works by flow cytometry). This metabolic uptake assay was developed by Linda Sinclair – Nature communications 2018 <https://www.nature.com/articles/s41467-018-04366-7>

*Again, it would be very nice to know which NK subpopulations uptake kynurenine.

We now include this data in supplementary figure 1 and below

Data from supplementary figure 1 showing the subset distribution (CD27 CD11b) of kynurenine⁺ NK cells.

*Authors demonstrated that NK cells increase kynurenine following FV infection but an experiment showing the direct effect of kynurenine on NK (function?) during infection is critical.

We are using kynurenine as a measure of Slc7a5 activity. We are not proposing that kynurenine is present during infection, nor that it is regulating NK cell function. See point and reference above.

-In Fig 5, a lot of experiments has to be done:

*It would be appreciated to include for each experiment Slc7a5^{NK-KO} at steady state.

We have done a thorough phenotyping of the Slc7a5-KO mice and at steady state and the NK cells are not significantly affected by Slc7a5 deficiency in terms of size and effector functions such as cytokine production. We include this data in supplemental figure 2 and below.

Data from supplementary figure 2 showing no differences between the Slc7a5^{NK-WT} and Slc7a5^{NK-KO} mice at steady state.

*In Fig 5e, authors showed that specific depletion of Slc7a5 leads to a decrease in NK cell numbers in BM and spleen. Once again, a histogram showing all NK subpopulations in order to know which NK subsets is affected by this depletion is missing.

We show the differences in the CD27⁺CD11b⁺ NK cell subsets at steady state in the manuscript (Figure 5b) and we are happy to provide the data for the NK cell subset distribution in Slc7a5 WT and KO mice during FV infection (below) but we are of the opinion that it does not add significantly to the manuscript. Similar to the CD27⁺CD11b⁺ subsets at steady state, we see an increase in CD27⁺CD11b⁻ NK cells and a decrease in CD27⁺CD11b⁺ NK cells whereas we do not detect significant differences in CD27⁻CD11b⁻ and CD27⁻CD11b⁺ NK cell subsets. Thus, we doubt that this figure would add any important information to the manuscript, but we are grateful for guidance by the reviewer and the editors.

Figure R4: Subset distribution of NK cells in *Slc7a5*^{NK-WT} and *Slc7a5*^{NK-KO} mice. *Slc7a5*^{NK-WT} and *Slc7a5*^{NK-KO} mice were infected with FV and spleen and bone marrow was harvested at 7 dpi. NK cells were analysed for CD27 and CD11b. Significant differences were analysed by an unpaired *t* test.

*Authors concluded that Slc7a5 transporter is dispensable for NK metabolism. The conclusion is dangerous given that authors have analyzed only few markers related to metabolism (CD71 and Myc). A Seahorse analysis of NK cells from Slc7a5NK-WT vs Slc7a5NK-KO at steady state and upon FV infection will reinforce this light conclusion. We did not have the numbers of KO mice required for seahorse analysis (5 mice required to pool cells for a single replicate) but we now include data on mitochondrial parameters, the

expression of CD98 and the uptake of transferrin to strengthen this argument. (Figure 5i and below)

Data from figure 5i are showing no differences between the $Slc7a5^{NK-WT}$ and $Slc7a5^{NK-KO}$ mice after FV infection for metabolism-related parameters.

-In Fig. 7, Littwitz-Salomon et al. show that iron deprivation inhibits NK cell responses against FV infection. However, an analysis of viral load during 7 days (as shown in Fig. 1a) between mice injected or not with mHep is missing and important.

Our previous analysis (Fig. 1b) has shown that day 7 post FV infection resulted in augmented NK cell activation. For this reason, day 7 was the most appropriate time point to investigate whether iron deficiency leads to elevated viral loads through inducing NK cell dysfunction. We also now show data strengthening the argument that low iron is having a direct effect on NK cells and so viral loads (see below).

Furthermore, iron is also uptake by other immune cells. How can we be sure that the mHep injection has a direct effect on NK?

We agree that iron is important for other immune cells and so we have addressed the question of whether the mHep injection is having a specific effect on the NK compartment in a number of ways.

(1) We considered whether there might be a defect in the production of the cytokines (IL-15 and IL-18), by macrophages and dendritic cells, that are known to activate NK cells in response to FV virus infection. While there were some changes in the activation of macrophages and dendritic cells in mice infected with FV and administered mHep, there was no decrease in the abundance of IL-15 or IL-18 levels measured in the serum and the spleens. In fact, there was a trend towards increased levels of IL-15 in the spleen of mHep treated mice and significantly increased levels of IL-12 and IL-18. (Figure 6j and below)

Data from figure 6j is showing the cytokine concentrations of IL-12, IL-15 and IL-18 in the spleen after FV infection and mHep treatment.

(2) Our previous work showed that if NK cells are depleted from the mouse (injection of NK1.1 antibody) there is an increase in FV infected cells - published in JV in 2017 doi: 10.1128/JVI.01122-17.

We reasoned if NK cells were dysfunctional in mHep treated mice following FV infection then removing NK cells would have no additional effect on viral loads. Indeed, when we depleted NK cells from the mice through the injection of anti-NK1.1 (clone PK136) prior to infection with FV + mHep treatment, mice lacking NK cells showed no increase in virally infected cells. (Figure 7i and j, and below). These data argue that NK cells require iron for their antiviral effector functions and explain the difference in viral loads in Figure 7.

Data from figure 7i and 7j showing the depletion efficiency of NK cells and the viral loads after mHep-treatment and NK cell depletion.

(3) We considered the direct impact of reduced iron availability on NK cell response. In these experiments NK cells were activated ex vivo in the presence of increasing concentrations of an iron chelator and the impact on NK cell growth (cell size, FSC), NK cell metabolism (mitochondrial mass) and NK cells function (perforin and granzyme B expression) investigated. All these parameters were reduced in a dose dependent manner and importantly were restored with the additional supplementation of the cultures with FeSO₄. (Figure 7e-h and below)

Data from figure 7 e-h. In vitro cultures of NK cells and treatment with the iron chelator DFO resulted in a dose dependent reduction of cell size, MitoTrackerGreen and Perforin as well as Granzyme B expression. The NK cell phenotype could be rescued by the addition of iron (FeSO₄).

We feel that taken together these new data make a strong argument that limited iron availability affects FV infection largely through the inhibition of NK cell metabolism and function.

Authors might show the direct effect of iron uptake on NK cells and their antiviral responses (in vitro experiments?)

We have looked at the impact of an iron chelator on NK cell response *in vitro* and added this data to figure 7 (see above).

-Finally, an experiment where the addition of kynurenine or iron (or both) ameliorates NK function and boosts their antiviral response is missing and required.

As discussed above, this study was not studying kynurenine in any way. Kynurenine uptake is simply an uptake assay to quantify Slc7a5 activity.

We have included some *in vitro* data showing that adding back ameliorates the negative impact of an iron chelator on metabolic and functional NK cell parameters (see above). Iron supplementation would only be expected to boost responses if we had evidence iron is significantly limiting *in vivo*, from our serum iron measurements on day 7 this is unlikely to be the case in murine FV infection. From our previous work further increasing iron availability to supraphysiological levels does not improve T cell responses *in vivo*. However high hepcidin and hypoferremia are observed in experimental norovirus infection in humans (Williams et al Am J Clin Nutr 2019) and hypoferremia occurs during acute viraemia of emerging HIV infection (Armitage et al PNAS 2014) and in severe COVID-19 (Shah et al Crit Care 2020). In these scenarios low serum iron could impact on the NK cell response: therefore hepcidin mediated hypoferremia is a relevant pathophysiological condition.

*Minor issues:

-In the introduction, references must be added (lines 48 and 62)

We have added references as suggested

-Line 163: “v” is missing for “NK cell activation”

Thank you for spotting this. We have now corrected this.

-In the 4th part of the text, the title “NK cells increase amino acid uptake following FV infection” should be replaced by “NK cells increase KYNURENINE uptake following FV infection” since no other amino acids have been shown in this part.

We apologize that we have not explained the kynurenine uptake assay good enough and add more information in the manuscript (lines 257-258). Kynurenine is just a proxy to measure amino acid uptake through slc7a5 – we have not studied the impact of kynurenine on NK cells (see Sinclair et al 2018, Nat Comms)

-In Fig 2, the authors wanted to know if acute FV infection induces metabolic changes in NK cells. Fig 2a-c have to be moved as Supplementary figures. It's out of context. The increase in size is not always correlated with metabolic changes.

In our experience of studying the metabolic changes that accompany lymphocyte activation, we would say that changes in cell size during this process are a rather good measure of the metabolic processes that are happening within those cells to mediate such changes in size – anabolic processes, nutrient uptake and cellular biosynthesis (see references below).

Amino acid-dependent cMyc expression is essential for NK cell metabolic and functional responses in mice.

Róisín M Loftus , Nadine Assmann 1, Nidhi Kedia-Mehta , Katie L O'Brien , Arianne Garcia , Conor Gillespie , Jens L Hukelmann , Peter J Oefner , Angus I Lamond , Clair M Gardiner , Katja Dettmer , Doreen A Cantrell , Linda V Sinclair , David K Finlay . Nat Commun. 2018 Jun 14;9(1):2341. doi: 10.1038/s41467-018-04719-2.

Srebp-controlled glucose metabolism is essential for NK cell functional responses

Nadine Assmann, Katie L O'Brien, Raymond P Donnelly, Lydia Dyck , Vanessa Zaiatz-Bittencourt , Róisín M Loftus , Paul Heinrich , Peter J Oefner , Lydia Lynch , Clair M Gardiner , Katja Dettmer , David K Finlay . Nat Immunol. 2017 Nov;18(11):1197-1206. doi: 10.1038/ni.3838. Epub 2017 Sep 18.

mTORC1-dependent metabolic reprogramming is a prerequisite for NK cell effector function
Raymond P Donnelly , Róisín M Loftus , Sinéad E Keating , Kevin T Liou , Christine A Biron , Clair M Gardiner , David K Finlay. J Immunol . 2014 Nov 1;193(9):4477-84. doi: 10.4049/jimmunol.1401558. Epub 2014 Sep 26.

-Fig 2: To really know if acute FV infection induces metabolic changes in NK cells, it would help to actually do a comparison of markers (CD98, CD71) between naïve vs FV mice and between the different subpopulations of NK cells.

As mentioned above, we feel that including every parameter studied for all the different subsets would make the manuscript difficult to navigate except for absolute NK cell experts. We want to publish in Nature Communications to reach a wide audience. We have separated some of these metabolic parameters in to the different subsets (below) and as you can see they invariably increase in all subsets. We can include these data in the supplementary figure if reviewer 3 deems it essential.

-Fig 5b axis labeling is missing

Thank you for spotting this. We have now corrected this.

-Fig 5h: In the paper from Loftus et al. Nat Com, 2018, the authors showed that Slca7a5 controls cMyc levels in NK cells. As control, it would be great to add the expression of cMyc in Slc7a5NK-WT vs Slc7a5NK-KO at steady state (as control).

In the steady state NK cells do not express cMyc and there is not difference in Slc7a5 KO mice. See supplementary figure 2b.

-Fig 6 and Fig 7 showing that iron availability is important for NK cells during infection and that iron deprivation inhibits NK response should be combined in a single figure.

We have added additional data to the figures, thus, we had to split it into two figures not to ask too much of the readers.

-Most graphs show NK frequency, whereas in fig 5e, 5f, histograms represent NK number. Frequencies in these figures or frequencies and numbers for every panels are required.

The reviewer is right, we are showing total NK cell numbers in Figure 5a, e, and f and provided percentages for 5f (KI-67) also in the supplementary figure 2c. In our hand, for analysing whole populations such as the population of NK cells, absolute numbers are more informative than frequencies because frequencies also depend on the flux/cell death/proliferation of other immune cell populations such as T cells.

REVIEWER COMMENTS

Reviewer #1 (Remarks to the Author):

The authors have sufficiently addressed my questions and concerns. The revised MS contains new data and discussion which was missing in its previous version.

Importantly, the authors show and discuss the implication on host outcome.

An important note: the authors refer to a Supplementary Figure 3 in the rebuttal letter, where no such figure exists in the manuscript. The MS does not refer to such a figure but it is important that the authors go over the text to make sure no such mistakes are made there.

Reviewer #2 (Remarks to the Author):

Manuscript Nr: NCOMMS-20-51020A

Littwitz-Salomon et al., "Metabolic requirements of NK cells during the response against retroviral infection"

The authors demonstrate that NK cells expand during Friend retrovirus (FV) infection and acquire metabolic changes that are consistent with NK cell activation. Compromising amino acid uptake by Slc7a5 does not seem to affect most NK cell effector functions except for cytotoxicity which is measured against YAC-1 and FV derived tumor cells. In contrast to this rather modest phenotype, hypoferrremia induced by mini-hepcidin affects both NK cell derived cytokine production and their cytotoxicity. This stronger inhibition of NK cell function is then also associated with elevated numbers of FV infected cells in the bone marrow. However, this effect might not be exclusively due to functional differences in the NK cell compartment.

In their revised manuscript version, the authors have addressed all of my previous concerns. Most importantly, the authors could show that NK cell depletion has no effect on viral infection during hypoferrremia. Therefore, the revised study seems to be significantly improved.

Reviewer #3 (Remarks to the Author):

Lots of work has been done. However, there are still concerns to be addressed as listed below:

1. In Fig 1c, only data related on infected mice have been included. Naive condition is missing; it would be nice to compare these subpopulations between naive and infected mice to see the changes during infection.
2. Fig 3, only data related on infected mice have been included. Naive condition is missing; it would be nice to compare these parameters between naive and infected mice to see the changes during infection.
3. Expression of Slc7a5 expression can be performed by qPCR even there is no available antibody. You used kynurenine which serves as measure of amino acid uptake through the Slc7a5 transporter and showed increased activity of the L-amino acid transporters in NK during FV infection. We know that Slc7a5 transports a lot of amino acids. That's why it would be nice and important to know which amino acids are preferentially uptake by NK cells during FV infection (metabolomic analysis for example).
4. Thank you for adding data on the impact of iron on NK cell responses. However, we still don't know the impact on the antiviral responses... I expected an analysis showing the FV-infected cells with the different concentration of iron chelator or by adding iron to correlate the impact of iron on NK and antiviral responses. Proposal for an experiment to do: pre-incubate NK cells with iron chelator or in contrast adding iron, and reinject them in FV-infected mice and then check the number of FV-infected cells. Here, it will be the proof that iron is important for NK anti viral response.
5. Some mistakes in the text:
 - line 164 "FV infection induces metabolically active NK cells"
 - line 209 "Mitochondria generates..."
 - line 219 "it induces the maximal polarization" - line 221 "it depolarises ..."

Please find below our point-by-point responses to the reviewers concerns in red text.

Reviewer #1 (Remarks to the Author):

The authors have sufficiently addressed my questions and concerns. The revised MS contains new data and discussion which was missing in its previous version.

Importantly, the authors show and discuss the implication on host outcome.

An important note: the authors refer to a Supplementary Figure 3 in the rebuttal letter, where no such figure exists in the manuscript. The MS does not refer to such a figure but it is important that the authors go over the text to make sure no such mistakes are made there.

We have revised our manuscript.

Reviewer #2 (Remarks to the Author):

Manuscript Nr: NCOMMS-20-51020A

Littwitz-Salomon et al., "Metabolic requirements of NK cells during the response against retroviral infection"

The authors demonstrate that NK cells expand during Friend retrovirus (FV) infection and acquire metabolic changes that are consistent with NK cell activation. Compromising amino acid uptake by Slc7a5 does not seem to affect most NK cell effector functions except for cytotoxicity which is measured against YAC-1 and FV derived tumor cells. In contrast to this rather modest phenotype, hypoferremia induced by mini-hepcidin affects both NK cell derived cytokine production and their cytotoxicity. This stronger inhibition of NK cell function is then also associated with elevated numbers of FV infected cells in the bone marrow. However, this effect might not be exclusively due to functional differences in the NK cell compartment.

In their revised manuscript version, the authors have addressed all of my previous concerns. Most importantly, the authors could show that NK cell depletion has no effect on viral infection during hypoferremia. Therefore, the revised study seems to be significantly improved.

Reviewer #3 (Remarks to the Author):

Lots of work has been done. However, there are still concerns to be addressed as listed below:

1. In Fig 1c, only data related on infected mice have been included. Naive condition is missing; it would be nice to compare these subpopulations between naive and infected mice to see the changes during infection.

We are a bit confused by this comment as in Figure 1c we have already done exactly as the reviewer is requesting and have compared naive and FV-infected mice. We have looked through the original comments from reviewer 3 to try and understand what this comment is in reference to but we are mystified. As Figure 1c stands we have compared NK cells from naïve mice and FV-infected mice in spleen and BM and labelled this clearly at the bottom of the graph.

2. Fig 3, only data related on infected mice have been included. Naive condition is missing; it would be nice to compare these parameters between naive and infected mice to see the changes during infection.

As with point 1, we are rather confused here too as figure 3 already compares naïve versus FV virus throughout. As it stands figure 3 a, c, d, e, g, h, l j, k, n, o compares the metabolic parameters of NK cells from naïve mice (white bars) and those infected with FV (grey bars). The histogram that is figure 3b compares NK cells from naïve mice (green) and those infected with FV (orange). The extracellular flux traces in figure 3f and 3i compare NK cells from naïve mice (white circles) and those infected with FV (grey squares). The only figure that does not have data for NK cells from naïve mice versus FV-infected mice is figure 3m, which is a figure that shows the validation of the kynurenine Slc7a5 uptake assay.

3. Expression of Slc7a5 expression can be performed by qPCR even there is no available antibody.

We agree that rtPCR could indeed be used to monitor the expression of Slc7a5 mRNA but we are of the firm opinion that demonstrating altered biological activity of Slc7a5 amino acid transport is much more important than measure than mRNA expression. Changes in mRNA (or lack thereof) do not necessarily reflect changes in protein expression and even protein expression may not reflect biological activity (e.g. should the transporter not be localized to the plasma membrane).

We have shown that in NK cells Slc7a5/Slc3a2 (i.e LAT1) is the only LAT transporter expressed (published in Nat Comms 2018 - Loftus et al.) and now herein we show of LAT activity is increased upon infection (increased kynurenine uptake + all the relevant controls) and this increased activity is completely lost when we delete Slc7a5 specifically in NK cells. Combined this tells us that Slc7a5 is the only LAT activity in NK cells during infection and that the biological activity of this transporter increases in NK cells responding to FV infection. We do not think that monitoring Slc7a5 mRNA would add significant value to this manuscript.

You used kynurenine which serves as measure of amino acid uptake through the Slc7a5 transporter and showed increased activity of the L-amino acid transporters in NK during FV infection. We know that Slc7a5 transports a lot of amino acids. That's why it would be nice and important to know which amino acids are preferentially uptake by NK cells during FV infection (metabolomic analysis for example).

Certainly, Slc7a5 will take up a number of different amino acids. What exact amino acids that travel through Slc7a5 at a given moment depends on (1) the KD of a given amino acid for the transporter (worked out previously by others that focus on amino acid transport kinetics) and (2) the concentrations of the all the potential Slc7a5 cargo amino acids in the external microenvironment. Therefore, to know what amino acids that an NK cell responding to FV infection is taking up would require us to do in vivo amino acid uptake assays. This is because if you do any metabolomics or ex vivo uptake assays your relative uptake of the different amino acids is determined by what amino acids and their concentrations are in the buffer that you put the cells into, and this does not reflect the situation in vivo at all. In vivo amino acid uptake assays are not possible with the technologies available to the field as a whole at the moment. Developing single cell amino acid uptake assays is something that we are working on with our ERC-CoG research programme but this project is still in the in vitro and ex vivo stage. Also, considering that Slc7a5 expression is not an important factor in determining FV infection outcomes, I would argue that knowing what amino acids are transported by Slc7a5 in this context is not crucial.

4. Thank you for adding data on the impact of iron on NK cell responses. However, we still don't know the impact on the antiviral responses... I expected an analysis showing the FV-infected cells with the different concentration of iron chelator or by adding iron to correlate the impact of iron on

NK and antiviral responses. Proposal for an experiment to do: pre-incubate NK cells with iron chelator or in contrast adding iron, and reinject them in FV-infected mice and then check the number of FV-infected cells. Here, it will be the proof that iron is important for NK anti viral response.

You are suggesting to treat NK cells with an iron chelator (like DFO) ex vivo and then adoptively transfer them into FV-infected mice to prove that iron depletion of NK cells directly affects NK cell antiviral responses and subsequently viral loads. We have discussed this with our collaborators Alexander Drakesmith and Joe Frost, who are experts in the field of iron and immunology, and we all agree that this experiment is highly unlikely to provide us with any additional meaningful data. As soon as the DFO-treated NK cells are transferred into mice, they would be in an iron replete environment and will take up as much iron as they want leading to the restoration of NK cell functionality.

We have discussed alternative experiments that might address reviewer 3's concerns but all the options involve the generation of complex transgenic mice (import, breeding and crossing would take around 1 year and a substantial financial commitment) and due to the complicated experimental designs required, the technical success of such experiments would still be uncertain. We hope reviewer 3 understands that there is no reasonable way to generate additional data to further confirm a direct effect of low iron on NK cell response to FV infection and is therefore satisfied with the evidence we have generated in the context of what all reviewer agree is an important body of research.

5. Some mistakes in the text:

-line 164 "FV infection induces metabolically active NK cells"

Thank you for this comment. We have changed it in the revised manuscript.

-line 209 "Mitochondria generates..."

Thank you for this suggestion. We are referring to multiple mitochondria that is the reason why we wrote "mitochondria generate chemical energy".

-line 219 "it induces the maximal polarization" - line 221 "it depolarises ..."

Thank you for spotting this out.